# *Bacillus thuringiensis* toxins divert progenitor cells toward enteroendocrine fate by decreasing cell adhesion with intestinal stem cells in *Drosophila*

Rouba Jneid[1,2†], Rihab Loudhaief[1‡], Nathalie Zucchini-Pascal[1], Marie-Paule Nawrot-Esposito[1], Arnaud Fichant[1,3], Raphael Rousset[1], Mathilde Bonis[3], Dani Osman[2§], Armel Gallet[1]*

[1]Universite Cote d'Azur, CNRS, INRAE, Sophia Antipolis, France; [2]Faculty of Sciences III and Azm Center for Research in Biotechnology and its Applications, LBA3B, EDST, Lebanese University, Tripoli, Lebanon; [3]Laboratory for Food Safety, University Paris-Est, French Agency for Food, Environmental and Occupational Health & Safety, Maisons-Alfort, France

*For correspondence:
gallet@unice.fr

Present address: †Department of Biology, Faculty of Science, University of Copenhagen, Copenhagen, Denmark; ‡K8520 IWK Health Centre 8th Floor East Research 5850/5980 University Avenue Halifax, Nova Scotia, Canada; §UMR PIMIT (Processus Infectieux en Milieu Insulaire Tropical) CNRS 9192- INSERM 1187-IRD 249-Université de La Réunion, île de La Réunion, France

Competing interest: The authors declare that no competing interests exist.

**Abstract** *Bacillus thuringiensis* subsp. *kurstaki* (*Btk*) is a strong pathogen toward lepidopteran larvae thanks to specific Cry toxins causing leaky gut phenotypes. Hence, *Btk* and its toxins are used worldwide as microbial insecticide and in genetically modified crops, respectively, to fight crop pests. However, *Btk* belongs to the *B. cereus* group, some strains of which are well known human opportunistic pathogens. Therefore, ingestion of *Btk* along with food may threaten organisms not susceptible to *Btk* infection. Here we show that Cry1A toxins induce enterocyte death and intestinal stem cell (ISC) proliferation in the midgut of *Drosophila melanogaster*, an organism non-susceptible to *Btk*. Surprisingly, a high proportion of the ISC daughter cells differentiate into enteroendocrine cells instead of their initial enterocyte destiny. We show that Cry1A toxins weaken the E-Cadherin-dependent adherens junction between the ISC and its immediate daughter progenitor, leading the latter to adopt an enteroendocrine fate. Hence, although not lethal to non-susceptible organisms, Cry toxins can interfere with conserved cell adhesion mechanisms, thereby disrupting intestinal homeostasis and endocrine functions.

## Editor's evaluation

The microbial pathogen Bacillus thuringiensis subsp. kurstaki (Btk) and its Cry toxins are used extensively to kill lepidopteran crop pests. Although Btk is not lethal to *Drosophila*, Jneid et al. present convincing evidence that Btk's Cry1A toxins disrupt *Drosophila* intestinal homeostasis by inducing enterocyte death, activating stem cell divisions, and promoting excess enteroendocrine differentiation via weakening of adherens junctions between stem cells and terminal daughter cells. These important findings raise the possibility that Btk and Cry-based insecticides may alter the intestinal lining of non-targeted animal species.

## Introduction

The gut lining is undergoing constant damage caused by environmental aggressors (pesticides, drugs, viruses, bacteria and toxins) ingested along with food. The gut quickly responds to these aggressions by accelerating its epithelium renewal to replace damaged cells. Over the past decade,

studies in *Drosophila melanogaster* have contributed substantially to the understanding of the cellular and molecular mechanisms controlling the maintenance of intestinal homeostasis and regeneration. These mechanisms have proven to be highly conserved in the animal kingdom. In *Drosophila*, resident intestinal stem cells (ISCs) are the guarantors of this cell renewal process. Under normal conditions, asymmetric division of an ISC gives rise to a new ISC (to maintain the pool of ISC) and to a daughter progenitor cell that can commit to two different paths of differentiation (*de Navascués et al., 2012*; *Goulas et al., 2012*; *O'Brien et al., 2011*; *Perdigoto et al., 2011*; *Tian and Jiang, 2014*). The enteroblasts (EBs) and enteroendocrine precursors (EEPs) are the precursors of enterocytes (ECs) and enteroendocrine cells (EEs), respectively (*Guo et al., 2021*; *Joly and Rousset, 2020*; *Pasco et al., 2015*). ECs are the main intestinal epithelial bricks constituting an efficient barrier against aggressors and are therefore their first victims. The damaged or dying ECs emit cytokines, which stimulate the proliferation of ISCs to augment the pool of EBs that will differentiate into ECs to replace the damaged ones (*Bonfini et al., 2016*; *Osman et al., 2012*). Two mechanisms underlying intestinal regeneration have been described. The first one is the 'cell renewal' model, which occurs under weak aggression conditions that does not induce EC apoptosis. In this case, ISC proliferation is low and the neo-EBs differentiate into ECs. However, this provokes a transient excess of ECs due to the absence of prior EC death. The gut cell homeostasis is subsequently reestablished by the removal of old ECs (*Loudhaief et al., 2017*). The second mechanism, called 'cell replenishment' or 'regenerative cell death', occurs after a strong aggression that induces massive EC apoptosis. In this case, a rapid ISC proliferation is followed by the differentiation of EBs into ECs to replace the dying ones (*Loudhaief et al., 2017*; *Vriz et al., 2014*) without producing supernumerary ECs.

*Bacillus thuringiensis* (*Bt*) bacteria are largely used as microbial insecticides to fight crop pests. *Bt* is a Gram-positive sporulating bacterium belonging to the *Bacillus cereus* (*Bc*) group (*Ehling-Schulz et al., 2019*). It was first identified and characterized for its specific entomopathogenic properties due to the presence of a crystal containing specific Cry protoxins, which are produced during the bacteria sporulation (*Rabinovitch et al., 2017*). Among all the subspecies of *Bt* inventoried (http://www.bgsc.org/), spores of *Bt* subsp. *Kurstaki* (*Btk*) are used to specifically kill lepidopteran larvae that threaten crops, through a cocktail of Cry toxins made of Cry1Aa, Cry1Ab, Cry1Ac, Cry2Aa and Cry2Ab (*Caballero et al., 2020*). Cry toxins sequentially bind to different receptors present in the midgut to exert their cytotoxicity. Among those receptors, the ones named Bt-R that belong to the Cadherin transmembrane cell adhesion molecules are primordial for the Cry1A holotype of toxins, allowing them to bind to enterocyte brush borders. The other receptors—such as Alkaline phosphatases, Aminopeptidases N, and ABC transporters—appear to account for the cytotoxicity that Cry exert toward susceptible organisms (*Adang et al., 2014*; *Gao et al., 2019*; *Li et al., 2020*; *Liu et al., 2018b*). In susceptible insects, upon ingestion of spores and crystals, the basic midgut pH dissolves the crystals, releasing the Cry protoxins. Then, digestive enzymes cleave Cry protoxins (130kD and 72kD for proCry1A and proCry2A, respectively) into activated Cry toxins (around 67kD) allowing them to bind their midgut receptors. Thereby, Cry toxins form pores in the plasma membrane of ECs, ultimately leading to their death. An alternative mode of action of Cry toxins suggested that Cry binding to Cadherin induces an intracellular flux of $Mg^{2+}$ resulting in EC apoptosis (*Castella et al., 2019*; *Mendoza-Almanza et al., 2020*). In both models, toxin-induced breaches within the gut lining allow bacteria (spores and vegetative cells) to reach the internal body cavity, generating a septicemia and subsequent death of the lepidopteran larvae within 2 or 3 days after ingestion of *Btk* spores (*Mendoza-Almanza et al., 2020*). It is assumed that *Btk* do not harm the intestine of non-susceptible organisms because, first, the intestinal pH is not suitable for the solubilization of the crystal of protoxins and, second, the Cry toxin receptors are absent from their gut epithelium (*Rubio-Infante and Moreno-Fierros, 2016*).

However, recent studies provide evidence that *Btk* also exhibits some adverse effects on non-susceptible organisms including humans. Indeed, *Bt* belongs to the *B. cereus* group to which some strains are well-known worldwide food-poisoning pathogens causing diarrheal-type illnesses (*Jovanovic et al., 2021*). Recently, *Bt* has also been implicated in foodborne outbreak events and the strains identified were indistinguishable from the commercial ones (*Biggel et al., 2021*; *Bonis et al., 2021*; *Johler et al., 2018*). Furthermore, we have shown that *Btk* spores and toxins at concentrations close to those recovered on vegetables after spraying induce growth defects and developmental delay in *Drosophila* larvae (*Nawrot-Esposito et al., 2020*). Increasing spore and toxin doses ultimately lead to larval lethality (*Babin et al., 2020*). Cry toxins produced by *Btk* are also used in genetically modified

crops (GMCs) (*ISAAA, 2017*), and it has been reported that GMC-produced Cry1Ab toxin is found in agricultural water stream networks at abnormally high doses that may affect the survival rate of non-susceptible insects (*Rosi-Marshall et al., 2007*). In a similar vein, laboratory studies have demonstrated genotoxic activity of Cry1Aa, Cry1Ab, Cry1Ac, and Cry2A in zebrafish rearing water (*Grisolia et al., 2009*). Based on all these data, our aim in this study was to decipher the interaction of *Btk* and its toxins with the intestinal epithelium using *Drosophila melanogaster*, an organism non-susceptible to *Btk* Cry toxin and a well-established model for studying host-pathogen interaction mechanisms.

Using environmental doses of spores and crystals of protoxins recovered on vegetables after treatment, we first showed that crystals of *Btk* Cry protoxins induced moderate enterocyte death that triggers a quick cell replenishment. We then demonstrated that the crystals diverted a higher number of progenitor cells from their initial EC fate toward an EE fate, generating an excess of EEs. Importantly, this effect was due to a weakened cell-cell interaction between ISC mother cells and progenitor daughter cells. We were able to rescue the crystal-dependent excess of EEs by specifically overexpressing the DE-Cadherin in ISC and progenitor daughters, reinforcing the strength of the adherens junction between these cells. Moreover, we found that among the five *Btk* Cry toxins, only the Cry1A holotype was able to induce this EE excess. Unexpectedly, we observed that *Btk* crystals are processed in the midgut of adult *Drosophila* as they are in that of susceptible-organisms, releasing activated Cry1A toxins. Hence, since our data demonstrate that Cry1A toxins disrupt conserved cellular processes, many non-susceptible organisms may exhibit an excess of EEs and consequently a disruption of their enteroendocrine functions.

## Results

### Crystals of *Btk* Cry protoxins induce EC death and stimulate proliferation of intestinal stem cells

During sporulation, *Btk* produces a crystal of protoxins that is lethal to lepidopteran larvae once ingested by the insect. To study the *Btk* effects on the non-susceptible organism *Drosophila melanogaster*, we orally infected flies with the SA11 *Btk* strain (hereafter named *Btk^SA11^*), which is widely utilized in commercial microbial insecticides. A suspension of spores/crystals in water was deposited on the fly medium corresponding to $10^6$ CFU (Colony Forming Unit) of spores per female for 4 cm². The impact on the gut of the spores alone, or the toxins alone, was also monitored using a *Btk* strain devoid of protoxin crystals (*Btk^ΔCry^*) or purified crystals, respectively (see Materials and methods).

The Gal4/UAS binary expression control system (*Brand and Perrimon, 1993*) allowed us to monitor first the effect of the spores/crystals on EC apoptosis by expressing the Caspase 3 sensor Casp::GFP; (*Schott et al., 2017*) under the control of the *myo1A-Gal4* EC driver (*myo1A>Casp::GFP*). With this transgenic combination, the GFP is detectable only when the Caspase 3 is activated in ECs. As a negative control, we fed flies with water alone. In all our experiments, we focused our observations on the posterior midgut (R4 region, https://flygut.epfl.ch/overview) (*Figure 1—figure supplement 1A*; *Buchon et al., 2013*) because this region is known to show a high stem cell renewal activity (*Marianes and Spradling, 2013*) and exhibits the strongest phenotypes (see below). One day post-ingestion, *Btk^SA11^* or purified crystals induced moderate apoptosis of the ECs compared to the control (*Figure 1A* and *Figure 1—figure supplement 1B*). However, the treatment with spores alone devoid of protoxin crystals (*Btk^ΔCry^*) did not induce EC death. Noteworthy, the overall morphology of the posterior midgut was not altered in the different conditions (*Figure 1—figure supplement 1A*).

Induction of cell death is known to strongly induce ISC proliferation in the whole midgut (*Biteau et al., 2008*; *Chatterjee and Ip, 2009*; *Jiang et al., 2009*; *Loudhaief et al., 2017*). This prompted us to assess the number of ISC mitoses in the different conditions, using an Anti-phospho-Histone H3 antibody marking mitotic cells. As expected, ISC mitotic indexes were stronger upon oral infection with *Btk^SA11^* spores or purified crystals than in the control (*Figure 1B*). In *Btk^ΔCry^* spore infection, mitotic figures were only moderately increased (*Figure 1B*). This is consistent with previous observations showing that a low dose ($10^6$ CFU per *Drosophila*) of *Btk* vegetative cells (that do not produce and contain crystals of Cry protoxins) only moderately activates ISC proliferation without inducing EC apoptosis (*Loudhaief et al., 2017*). We confirmed this increase in ISC proliferation by analyzing ISC density and proportion in the posterior midgut (R4 region). To specifically mark ISCs, we expressed the GFP under the control of the *Dl-Gal4* driver that is expressed in ISCs and EEPs (*Joly*

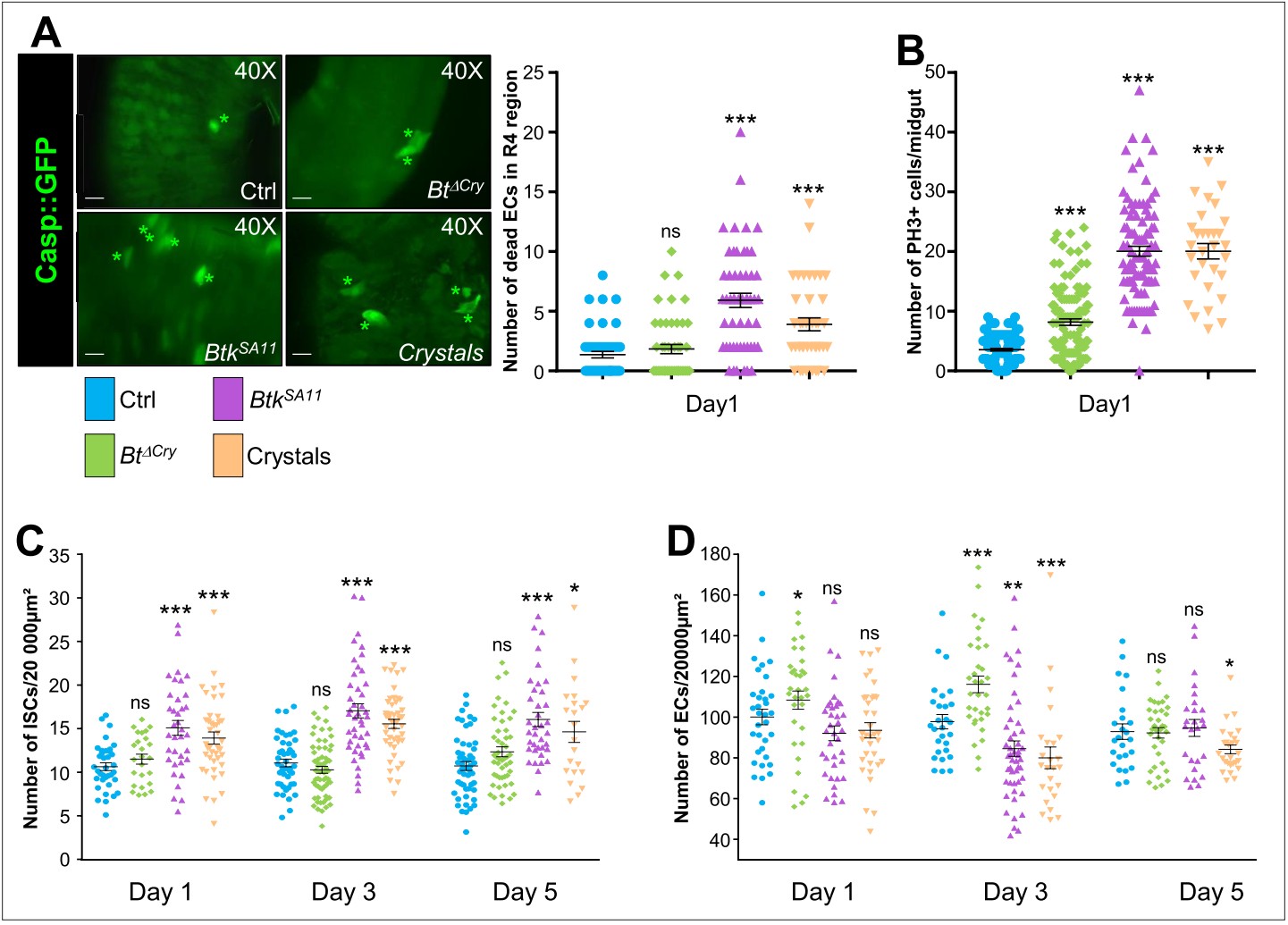

**Figure 1.** Crystals of *Btk* Cry protoxins induce EC death and stimulate proliferation of intestinal stem cells. (**A**) EC apoptosis was monitored by expressing the Caspase 3 sensor (Casp:: GFP) using the *myo1A-GAL4* EC driver (*myo1A>Casp::GFP*). With this transgenic combination, the GFP is detectable only when the Caspase 3 is activated in ECs. Left panel: ×40 magnification of a R4 subregion. Green stars mark GFP-positive dying ECs. Scale bar = 20 μm. Right panel: quantification of dead ECs 24 hr post ingestion (PI) in the posterior midgut (R4 region). (**B**) Quantification of mitoses using the anti-PH3 antibody in the whole midgut 24 hr PI. (**C**) ISC density in the R4 region of *esg >GFP* flies 24, 72, and 120 hr PI. (**D**) EC density in the R4 region of *myo1A>GFP* flies 24, 72, and 120 hr PI. Data is reported as mean ± SEM. ns = not significant; * (p≤0.05); ** (p≤0.01), *** (p≤0.001).

The online version of this article includes the following source data and figure supplement(s) for figure 1:

**Source data 1.** Cell type counting.

**Figure supplement 1.** Crystals of *Btk* Cry protoxins disturb intestinal homeostasis.

**Figure supplement 2.** Cell ratio analysis in the R4 region.

**Figure supplement 2—source data 1.** Cell ratio analysis in the R4 region.

---

*and Rousset, 2020*), and we co-stained with an anti-Prospero (Pros), an EEP and EE marker (ISCs were therefore GFP+, Pros-). While we observed an increase in ISC number with *Btk^SA11* spores or purified crystals (both density and ratio increased), *Btk^ΔCry* spores did not induce any increase (*Figure 1C* and *Figure 1—figure supplement 2A–B*). This could be explained by the fact that the moderate stimulation of ISC proliferation by *Btk^ΔCry* spores was not sufficient to promote a detectable increase in global ISC number. Nonetheless, to verify that ISC daughter cells committed to a process of differentiation upon *Btk^ΔCry* or *Btk^SA11* spore ingestion, we used the ReDDM *Drosophila* genetic tool (*Antonello et al., 2015*) under the control of the *Dl-Gal4* driver (*Dl*-ReDDM). This tool allows us to follow the progeny of the ISCs because they express a stable RFP (H2B::RFP) while the mother cells (the ISCs) express a labile GFP. After shifting the flies to 29 °C to activate the *Dl*-ReDDM tool, the GFP was only expressed

in Dl + cells (i.e. the ISCs and EEPs) while the H2B::RFP was expressed in Dl + cells but also stably transmitted to the progeny. We also used an anti-Pros to label EEPs and EEs. Consequently, ISCs were recognized by their expression of both GFP and RFP; EEPs expressed GFP, RFP and Pros, EEs expressed only Pros, EBs and ECs expressed only the RFP. As expected, both *Btk*^ΔCry^ or *Btk*^SA11^ spore ingestion promoted ISC daughter cell differentiation (*Figure 1—figure supplement 1C*) but surprisingly we observed an abnormal elevated number of EEP doublets upon ingestion of *Btk*^SA11^ spore (see below).

We next monitored the density and ratio of ECs using the fly strain *myo1A>GFP* allowing the expression of GFP in all ECs. *Btk*^ΔCry^ spores induced an increase of EC density at days 1 and 3 post-ingestion though their ratio was not altered; the right density of ECs was recovered 5 days after ingestion (*Figure 1D* and *Figure 1—figure supplements 1D and 2A, C*). Along with the low number of dying EC (*Figure 1A*) and the moderate induction of ISC proliferation (*Figure 1B*), our data strongly suggest that *Btk*^ΔCry^ spores weakly damage the intestinal epithelium, reminiscent of the 'cell renewal' process previously described after infection with poorly virulent bacteria (*Loudhaief et al., 2017*). On the contrary, ingestion of the *Btk*^SA11^ spores or purified crystals provoked a decrease in total EC number 3 days post-ingestion (both density and ratio dropped down) (*Figure 1D* and *Figure 1—figure supplements 1D and 2A, C*) that we attributed to EC apoptosis (*Figure 1A*; *Loudhaief et al., 2017*). A normal number of ECs was restored 5 days post-ingestion for *Btk*^SA11^ spores and to a lesser extent for purified crystals (*Figure 1D* and *Figure 1—figure supplements 1D and 2A, C*). Hence, *Btk*^SA11^ spores or purified crystals launch a process of regenerative cell death, inducing a strong proliferation of ISCs to quickly replenish the gut lining as previously described for strong pathogens (*Vriz et al., 2014*). Importantly, the ingestion of purified crystals containing the *Btk* Cry-protoxins recapitulates the midgut phenotypes caused by the ingestion of *Btk*^SA11^ spores.

## *Btk*^SA11^ spores induced an increase in EB, EEP and EE numbers

During the course of our experiments we observed that many more EEs were apparently present in the flies that had ingested *Btk*^SA11^ spores compared with the control flies (*Figure 2—figure supplement 1A–C*). As ECs derive from EBs and EEs from EEPs, we assessed the amount and the identity of the precursors in the different conditions of infection (H₂O; *Btk*^ΔCry^ spores, *Btk*^SA11^ spores and purified crystals). To count the EBs, we used a *Gal4* strain of *Drosophila* driving GFP expression specifically in EBs (*Su(H)>CD8::GFP*). As expected, a significant increase in the number of EBs was observed between the first and the fifth day after ingestion of *Btk*^ΔCry^, *Btk*^SA11^ or purified crystals (*Figure 2A* and *Figure 1—figure supplement 2A, D*). The EEP density was assessed using two markers: the GFP expressed in ISCs and progenitors (EBs and EEPs) using the *esg-Gal4* driver (*esg >GFP*) and a Pros staining which labels EEPs and EEs. Cells expressing both GFP and Pros corresponded to EEPs. While *Btk*^ΔCry^ spores did not modify the density of EEPs, ingestion of either *Btk*^SA11^ spores or purified crystals resulted in an increase in EEPs from day one onwards (*Figure 2B and D*). Since EEPs must differentiate into EEs, we then counted the differentiated EEs that were GFP-/Pros+. No difference in EE density was obtained with *Btk*^ΔCry^ spores compared to the control, whereas there was a net increase with the *Btk*^SA11^ spores and with the purified crystals (*Figure 2C and D*, *Figure 1—figure supplement 2A* and E, and *Figure 2—figure supplement 1A–D*). Interestingly, although this event was rare, we also observed that EEPs could undergo a cycle of mitosis that might contribute to the increase in EEPs and EEs (*Figure 2E*; *Biteau and Jasper, 2014*; *Li et al., 2014*; *Zeng and Hou, 2015*). Noticeably, we never observed such an increase in EEPs and EEs in the anterior part of the midgut (*Figure 2—figure supplement 1E–I*). Altogether, our results showed that the protoxin crystals of *Btk*^SA11^ were responsible for the excess of EEs in the posterior midgut.

## EE excess arises from newborn EEPs after ingestion of crystals of *Btk*^SA11^ protoxins

To demonstrate that the excess of EEPs and EEs arose from proliferating ISCs caused by the ingestion of protoxin crystals, we used the ReDDM cell lineage tracing system using the *esg-Gal4* driver (*esg-ReDDM* flies). We chose to analyze the progeny at day 3 post-ingestion (*Figure 3A*), when the increase in EEs reached its peak (*Figure 2C* and *Figure 1—figure supplement 1E*). According to the expression of specific cell markers and nucleus size, we could identify different cell types either that existed before the ingestion or that appeared after the ingestion of *Btk*^ΔCry^ spores, *Btk*^SA11^ spores or

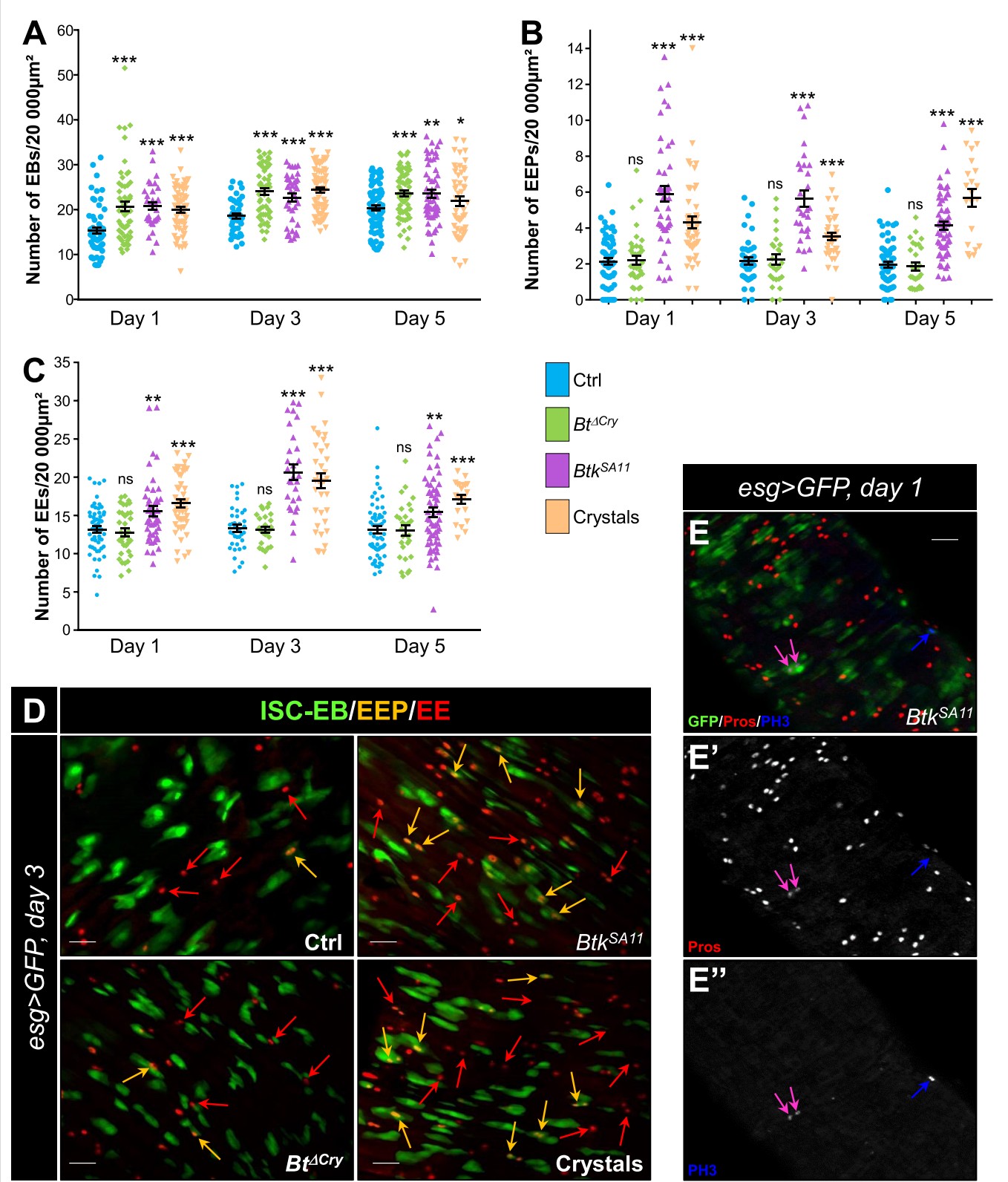

**Figure 2.** *Btk^SA11* spores induce an increase in EB, EEP and EE numbers. (**A–D**) Flies were fed with water, *Bt^ΔCry* spores, *Btk^SA11* spores or Crystals. (**E-E"**) Flies were fed with *Btk^SA11* spores. (**A–C**) Control (water-ctrl): blue; *Bt^ΔCry* spores: green; *Btk^SA11* spores: purple; Crystals: beige. (**A**) EB density in the R4 region of *Su(H)>CD8::GFP* flies 24, 48, and 72 h PI. (**B and C**) EEP (**B**) and EE (**C**) density in the R4 region of *esg >GFP* flies 24, 48, and 72 hr PI. (**D-E"**) R4 region of *esg >GFP* flies labeled with anti-Pros (Red). GFP was expressed in ISCs, EBs and EEPs, and Pros was expressed in EEPs (yellow arrows

*Figure 2 continued on next page*

*Figure 2 continued*

in D) and EEs (red arrows in D). (**E-E''**) PH3 staining (blue) marks mitosis. Pink arrows point to dividing EEPs and blue arrows point to dividing ISCs. ×40 magnification. Scale bar = 20 μm. Data is reported as mean ± SEM. ns = not significant; * (p≤0.05); ** (p≤0.01), *** (p≤0.001).

The online version of this article includes the following source data and figure supplement(s) for figure 2:

**Source data 1.** Cell type counting.

**Figure supplement 1.** *Btk^SA11* crystals induce an increase in EEP and EE number in the posterior midgut.

**Figure supplement 1—source data 1.** Cell counting.

purified crystals. Identities of the different cell types were defined as follows: ISCs and EBs were GFP+/RFP+/DAPI +with small nuclei (although EBs were bigger cell than ISCs); EEPs were GFP+/RFP+/Pros+/DAPI+; new EEs were RFP+/Pros+/DAPI+; old EEs were Pros+/DAPI+, new ECs were RFP+/DAPI + with big polyploid nuclei and old ECs were DAPI + with very big nuclei (*Figure 3B–E"*). In the control experiments, a few newborn ECs (red arrows in *Figure 3B and B'*, and *Figure 3G*) and rare newborn EEs (*Figure 3F*) appeared 3 days post-ingestion, reflecting the relative steady state of the cellular homeostasis. As expected for poorly virulent bacteria, ingestion of *Btk^ΔCry* spores induced the appearance of newborn ECs (red arrows in *Figure 3C and C'* and *Figure 3G*) and only rare newborn EEs resulted (pink arrow in *Figure 3C–C"* and *Figure 3F*). Similarly, ingestion of *Btk^SA11* spores or purified crystals promoted the appearance of newborn ECs (red arrows in *Figure 3D, D', E and E'* and *Figure 3G*) but, strikingly, a high number of newborn EEs appeared (pink arrows in *Figure 3D–E"* and *Figure 3F*). However, we could not rule out the possibility that the *Btk^SA11* spores altered EB behavior, pushing them toward an EE fate. To verify this possibility, we carried out a ReDDM lineage tracing using the *Su(H)-Gal 4* driver that is specifically expressed in EBs (*Su(H)-ReDDM* flies). In this case, no newborn EEs were detectable upon ingestion of the *Btk^SA11* spores while many newborn ECs were present (*Figure 3—figure supplement 1A–D*), indicating that EBs are not the source of the increase in the number of EEs. These data confirm previous observations that EEs never develop from Su(H)+EBs (*Biteau and Jasper, 2014*; *Zeng and Hou, 2015*).

Altogether, our data demonstrate that ingestion of *Btk^SA11* spores damages the intestinal epithelium, stimulating ISC proliferation. However, some of the progenitors make the choice to commit to an EEP/EE fate instead of an EB/EC fate. Consequently, there is a lack of new ECs to replace the dying ones and there is an excess of EEs. Interestingly, the crystals containing the Cry protoxins can recapitulate all the *Btk^SA11* spore effects. In contrast, the effect of *Btk^ΔCry* spores is less damaging for the gut epithelium. In this case, ISC proliferation is only weakly stimulated and the progenitors make the choice to commit to the EB/EC fate to replace the damaged ones.

## Crystals of *Btk^SA11* protoxins decrease ISC-progenitor cell-cell adhesion

It is well established that the Notch (N) signaling pathway governs progenitor differentiation and cell lineage choice in the adult *Drosophila* midgut (*Micchelli and Perrimon, 2006*; *Ohlstein and Spradling, 2006*; *Ohlstein and Spradling, 2007*; *Pasco et al., 2015*). Indeed, the transmembrane ligand Delta (Dl) expressed in ISCs binds to its N receptor present on the surface of progenitors. This induces the cleavage of the intracellular domain of N and its relocation into the nucleus to activate its target genes (*Perdigoto and Bardin, 2013*). Upon N activation, progenitors differentiate into EBs and then into ECs while in the absence/weak activation of N signaling, progenitors commit to an EEP/EE fate (*Beehler-Evans and Micchelli, 2015*; *Guo and Ohlstein, 2015*; *Ohlstein and Spradling, 2007*; *Sallé et al., 2017*). A prolonged and/or strong interaction between the ISC and its progenitor is necessary to reach the threshold of the N signaling activation sufficient to commit the progenitor to the EB/EC fate. A shorter and/or weaker interaction between the ISC and its progenitor weakly induces the N pathway, pushing the progenitor towards the EE fate (*Guisoni et al., 2017*; *Sallé et al., 2017*). The adherens junctions between ISCs and progenitors formed by E-Cadherins and Connectins intervene to prolong the contact between Dl and N, favoring the EB/EC fate (*Choi et al., 2011*; *Falo-Sanjuan and Bray, 2021*; *Maeda et al., 2008*; *Zhai et al., 2017*). As the consensus receptors for Cry1A toxins in target organisms are members of the Cadherin family (*Adang et al., 2014*), we wondered whether the *Btk^SA11* could interfere with the function of the adherens junctions. We hypothesized that ISC-progenitor interaction could be reduced via interference of Cry toxins with Cadherins, modifying the progenitor cell fate and thus explaining the excess in EE number seen after ingestion of *Btk^SA11* spores

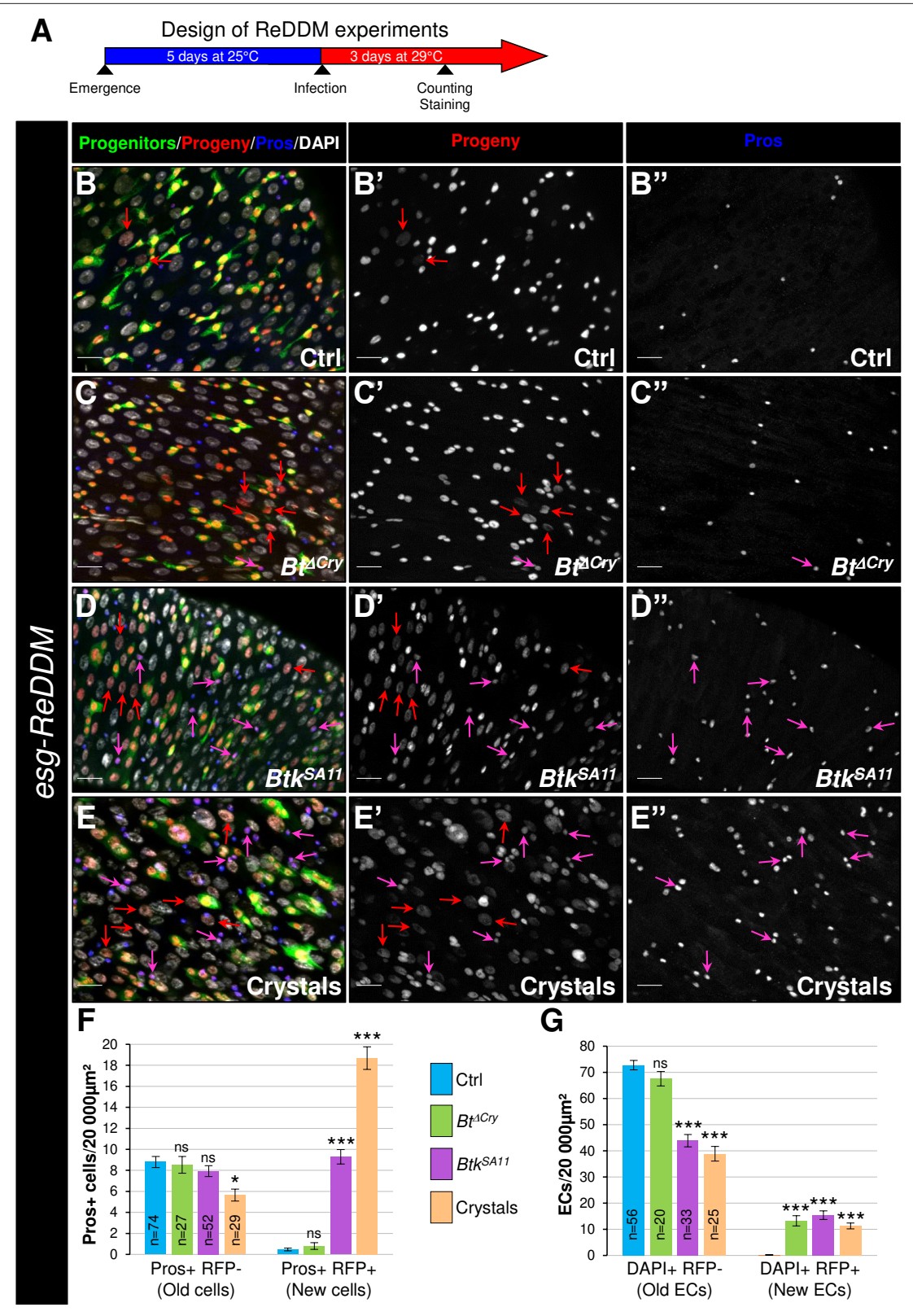

**Figure 3.** EE excess arises from newborn EEPs after ingestion of *Btk^SA11* crystals. (**A**) Schema of the experimental design for the *esg-ReDDM* cell lineage used in this entire figure. (**B-E''**) R4 region of *esg-ReDDM* flies. Midguts were stained for Pros (blue) and DAPI which marks nuclei (white in **B**, **C, D and E**). (**B-E''**) show the different cell types which either existed before the ingestion (green and red) or arise after the ingestion (red only) of water (B-B'', Ctrl), *Btk^ΔCry* spores (**C-C''**) or *Btk^SA11* spores (**D-D''**) and Crystals (**E-E''**). ISCs were GFP + RFP + DAPI +with small nuclei; EBs were GFP + RFP +

*Figure 3 continued on next page*

Figure 3 continued

DAPI +with bigger nuclei; EEPs were GFP + RFP + Pros +DAPI + ; new EEs were RFP+, Pros + DAPI + ; old EEs were Pros +DAPI + ; new ECs were RFP + DAPI + with polyploid big nuclei and old ECs were DAPI +with very big nuclei. 40 X magnification. Scale bar = 20 μm. (**F**) Counting old EEs (Pros + RFP-) and new EEs (Pros + RFP + ) in the conditions described in (**B–E**). (**G**) Counting old ECs (DAPI+) and new ECs (DAPI + RFP + ) in the conditions described in (**B–E**) n=number of 40 x images analyzed Data is reported as mean ± SEM. ns = not significant; * (p≤0.05); *** (p≤0.001).

The online version of this article includes the following source data and figure supplement(s) for figure 3:

**Source data 1.** Cell type counting.

**Figure supplement 1.** EBs do not give birth to EEs.

**Figure supplement 1—source data 1.** Cell type counting.

or purified crystals. To test this hypothesis, we labelled the intestines of *esg >GFP Drosophila* fed with *Btk^{ΔCry}* spores, *Btk^{SA11}* spores or purified crystals with the anti-Armadillo (Arm)/β-catenin antibody that strongly marks the adherens junctions. We observed an intense labeling at the level of the junctions between pairs of GFP + cells in the control (*Figure 4—figure supplement 1A–A", E*) or following intoxication by *Btk^{ΔCry}* spores (*Figure 4—figure supplement 1B–B, E*). Strikingly, this labeling became less intense following ingestion of *Btk^{SA11}* spores or purified crystals (*Figure 4—figure supplement 1C–E*), and correlated with an increase in EEPs number. To verify that weakening of cell junctions corresponded to cell shift towards an EEP fate, we used a *Drosophila* line expressing the endogenous DE-Cadherin (DE-Cad) fused to a Tomato tag in which progenitor cells (ISCs, EBs and EEPs) were labelled with GFP, while EEPs and EEs were marked with Pros (*Figure 4*). Using this genetic background, we first confirmed the decrease in the proportion of cells showing strong junctions between pairs of GFP + cells upon feeding with *Btk^{SA11}* spores or purified crystals (*Figure 4E*). Second, as expected, we observed that the weak Tomato::DE-Cad labelling between a GFP + cell (an ISC or an EB) and its neighboring cells was correlated with the expression of the EEP marker Pros (blue stars in *Figure 4A–D'* and *Table 1*). Together, our results suggested that crystals of Cry protoxins produced by *Btk^{SA11}* are responsible for the increase in the number of EEPs/EEs, this effect being associated with a decrease in intercellular adhesion between ISCs and progenitors.

## Increasing adherens junction strength rescues crystal-dependent cell fate diversion

To confirm that the crystals of *Btk^{SA11}* interfered with progenitor fate by disturbing adherens junctions, we wondered whether increasing the strength of cell adhesion between ISCs and progenitors could rescue the right number of EEPs/EEs. Thus, we overexpressed the DE-Cad in these cells using the *esg-ReDDM* flies (*Figure 5A–G* and *Figure 5—figure supplement 1A–E*). We analyzed the identity of newborn cells 3 days after ingestion of water (control), *Btk^{ΔCry}* or *Btk^{SA11}* spores, or purified crystals. First of all, we verified that DE-Cad overexpression in ISCs and progenitor cells (*esg + cells*) did not blocked ISC proliferation (*Figure 5—figure supplement 1E* compared to *Figure 3—figure supplement 1E*). Interestingly, in flies overexpressing the DE-Cad fed with *Btk^{SA11}* spores or purified crystals, we observed a rescue in the number of EEs (blue arrows in *Figure 5A–D'*, and *Figure 5E* compared to *Figure 3F*). In agreement, the number of ISC-progenitor pairs with a strong interaction was increased (*Figure 5G* compared to *Figure 4E*). Furthermore, as expected, more newborn ECs appeared (red arrows in *Figure 5B–D'* and *Figure 5F* compared to *Figure 3G*), strongly suggesting that increasing the cell adhesion between ISCs and progenitors rescued the progenitor fate disturbance generated by the *Btk^{SA11}* crystals. Surprisingly, overexpressing the Connectin, another cell adhesion molecule, which mediates hemophilic cell-cell adhesion (*Zhai et al., 2017*), in both ISCs and progenitors did not rescue the number of EEs following feeding with *Btk^{SA11}* spores or purified crystals (*Figure 5—figure supplement 2*). The emergence of new Pros + cells were still considerable upon ingestion of *Btk^{SA11}* or crystals at the expense of new ECs (*Figure 5—figure supplement 2I* and J). Of note, the overexpression of the Connectin in *esg+* cells did not impact ISC proliferation (*Figure 5—figure supplement 2K*). To confirm the disturbance of DE-Cad-dependent cell adhesion by purified crystals, we carried out an aggregation assay in *Drosophila* S2 cell culture. Indeed, S2 cells do not endogenously express the DE-Cad and display only a weak cell-cell adhesion phenotype (*Toret et al., 2014*). Transfection of S2 cells with a plasmid encoding a DE-Cad::GFP fusion resulted in large aggregate formation as early as 1 hr post-agitation (Ctrl in *Figure 5H* and *Figure 5—figure supplement 1F*). Adding purified crystals

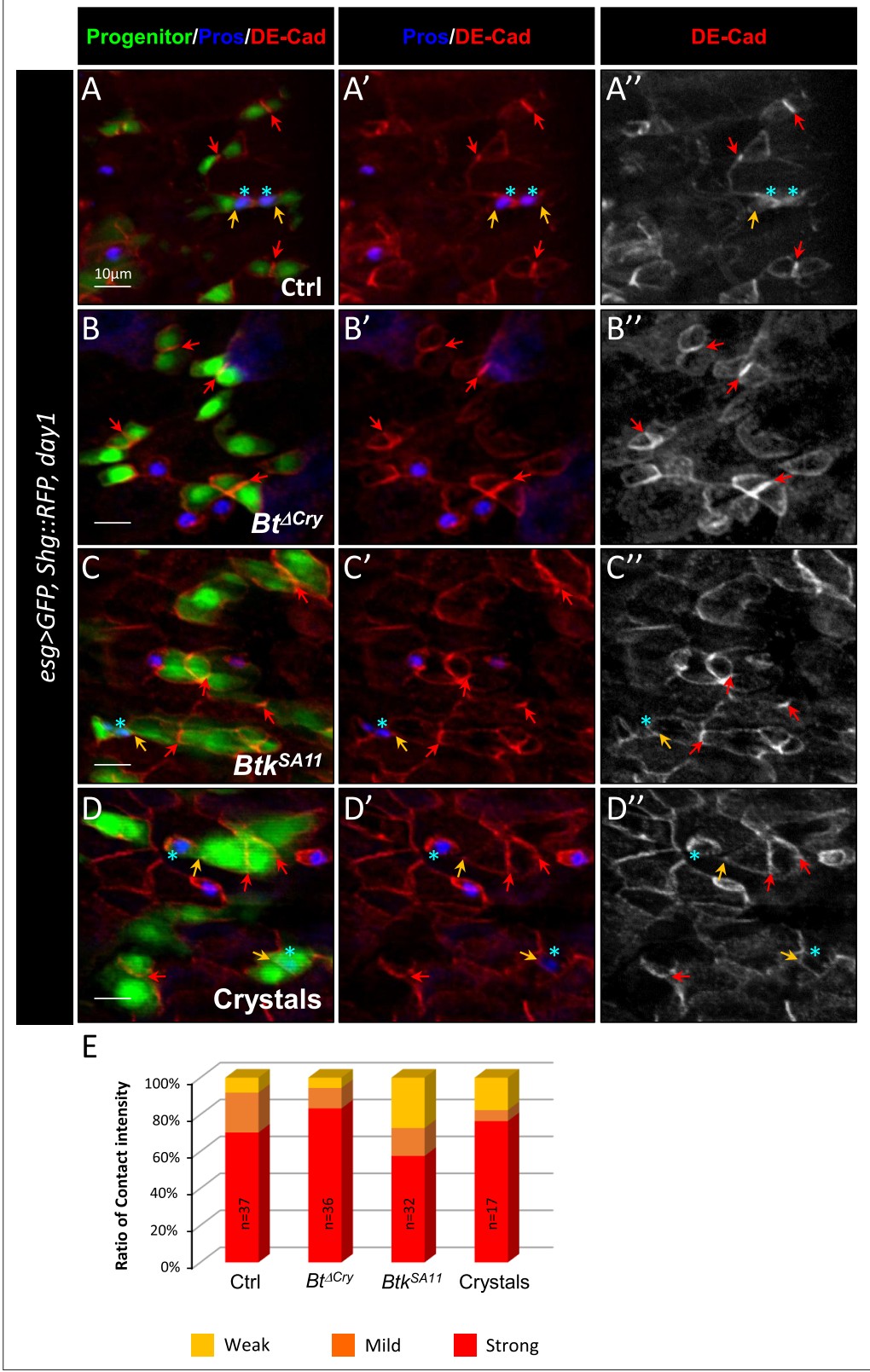

**Figure 4.** Btk crystals decrease ISC-Progenitor cell-cell adhesion. (**A–E**) *esg >UAS GFP, Tomato::shg Drosophila* midgut R4 region 24 hr PI of water (A-A" Ctrl), *Btk^{ΔCry}* spores (**B-B"**), *Btk^{SA11}* spores (**C-C"**) or crystals (**D-D"**). (**A-D"**) Midguts are labelled for Pros (blue), DE-Cadherin (red) and ISCs and progenitors (green). Red arrows point to the high intensity of adherens junctions staining between ISC and progenitors. Yellow arrows point to the weak

*Figure 4 continued on next page*

*Figure 4 continued*

intensity of adherens junction staining. Note that the high intensity of adherens junction staining is associated with ISC/EB interaction while the weak intensity of adherens junction staining is associated with ISC/EEP interaction (blue stars mark EEPs). ×40 magnification. Scale bar = 10 µm. (**E**) Graph representing the percentage of the different categories of cell contact intensity between ISCs and progenitors. n=number of cell pairs analyzed. Weak = Contact Intensity/Membrane Intensity <1.4; Mild = 1.4 < Contact Intensity/Membrane Intensity <1.6; Strong = Contact Intensity/Membrane Intensity >1.6.

The online version of this article includes the following source data and figure supplement(s) for figure 4:

**Source data 1.** Ratio of contact intensity.

**Figure supplement 1.** Btk bioinsecticide decreases ISC-Progenitor cell-cell adhesion.

**Figure supplement 1—source data 1.** Ratio of contact intensity.

to the cell culture medium strongly reduced the size of aggregates. Interestingly, purified Cry1Ab and Cry1Ac protoxins have the same effect although Cry1Ab needed a longer time to reduce the size of S2 cell aggregates (*Figure 5H–H"* and *Figure 5—figure supplement 1F*). Because Cadherins serve as receptors for the Cry1A toxin family in target Lepidoptera (*Adang et al., 2014*), our data suggest that in non-target organisms such as *Drosophila melanogaster*, Cry1A toxins could interfere physically with the well-conserved E-Cadherin.

## Cry1A toxins mimic *Btk^SA11^* spore effects

*Btk^SA11^* produces five different Cry toxins (Cry1Aa, Cry1Ab, Cry1Ac, Cry2Aa, and Cry2Ab) (*Caballero et al., 2020*). We investigated whether the increase in EEP/EE number was due to all toxins present in the crystals or to only one family of toxins (i.e. the Cry1A or Cry2A family). We made use of the *Btk^Cry1Ac^* strain (referred to as 4D4 in https://bgsc.org/) which produces crystals composed only of the Cry1Ac protoxin. We fed *esg >GFP* flies either with spores of *Btk^Cry1Ac^* or with purified Cry1Ac crystals. In both conditions, we observed a significant increase in the number of EEPs and EEs 3 days post-ingestion compared to controls (*Figure 6A* and *Figure 6—figure supplement 1A, C, D*). To verify whether other toxins of the Cry1A family induced a similar rise in EEP/EE number, we generated a *Btk^Cry1Ab^* strain (see Material and Methods) producing only the Cry1Ab toxins (*Figure 6—figure supplement 1I*). Similar to the *Btk^Cry1Ac^* spores, *Btk^Cry1Ab^* spores induced an increase in EEP/EE numbers (*Figure 6B* and *Figure 6—figure supplement 1B*). Unfortunately, no *Btk* strain yielding only Cry2A-containing crystals was available and we were unsuccessful in generating one. To overcome this, we used heterologous expression of Cry toxins in *E coli*. We first checked whether Cry1Ac protoxin produced and purified from *Escherichia coli* (*E. coli*) was indeed able to induce an increase in EEP/EE numbers. We also forced the activation of the Cry1Ac protoxin into an activated form in vitro (see Materials and methods). Interestingly, both the protoxin and the activated form of Cry1Ac were able to induce the expected phenotype, although the activated Cry1Ac form was more efficient (*Figure 6C* and *Figure 6—figure supplement 1E, F*). Conversely, both Cry2Aa protoxin and its activated form purified from *E coli* were unable to increase the number of EEP/EEs (*Figure 6C* and *Figure 6—figure supplement 1G and H*). Therefore, our data demonstrate that ingestion of Cry1A toxins was sufficient to induce a rise in the numbers of both EEPs and EEs.

## Cry1A Protoxins from *Btk* crystals are activated in the *Drosophila* midgut

Our data above showed that purified activated Cry1Ac toxin was more efficient for inducing an EEP/EE excess than the purified Cry1Ac protoxin. Interestingly, the magnitude of EEP/EE excess was similar using either Cry1Ac crystals or purified activated Cry1Ac toxin (compare *Figure 6A and C*), suggesting that Cry1Ac protoxins contained in the crystals were activated in the *Drosophila* intestine. However, the admitted model proposes that protoxins can be activated in vivo only in the intestine of the susceptible lepidopteran owing to the presence of appropriate digestive proteases specifically functioning at the basic pH and reducing conditions encountered in the larval midgut of lepidopteran (*Pardo-López et al., 2013*; *Soberón et al., 2009*; *Vachon et al., 2012*). We therefore wondered whether the effects we observed in vivo were due to the crystal on its own (i.e. protoxins) or to the activated Cry toxins after processing in the *Drosophila* midgut. We first monitored by western Blot the

**Table 1.** Cell junction intensity ratio measurement between pairs of progenitors. Cadherin::RFP labeling intensity were measured first at the cell junction between pairs of progenitors and second around the rest of the cell membrane (see *Figure 4—figure supplement 1F–G*). Ratio correspond to the Junction intensity/the rest of the membrane. Prospero positive progenitors were both GFP+/Pros+ (see *Figure 4*). Yellow highlight labels Pros + progenitors with a weak intensity ratio. Orange highlight labels Pros + progenitors with a medium intensity ratio. Red highlight labels Pros + progenitors with a strong intensity ratio.

| H2O Intensity ratio | Pros + | $Btk^{\Delta Cry}$ Intensity ratio | Pros + | $Btk^{SA11}$ Intensity ratio | Pros + | Crystals Intensity ratio | Pros + |
|---|---|---|---|---|---|---|---|
| 2,29 | | 1,46 | | 1,49 | Yes | 1,27 | Yes |
| 1,37 | | 2,27 | | 1,38 | Yes | 2,03 | |
| 1,43 | | 2,25 | | 1,91 | | 0,78 | Yes |
| 1,75 | | 1,92 | | 1,52 | | 1,18 | Yes |
| 1,87 | | 1,51 | | 1,89 | | 1,89 | |
| 2,37 | | 2,19 | | 1,88 | | 2,56 | |
| 2,54 | | 1,96 | | 1,19 | Yes | 2,20 | |
| 1,77 | | 1,91 | | 2,27 | | 3,12 | |
| 1,45 | | 1,35 | Yes | 1,63 | | 4,05 | |
| 1,49 | | 1,94 | | 1,39 | | 1,91 | |
| 1,83 | | 2,46 | | 1,54 | Yes | 1,70 | |
| 1,76 | | 2,93 | | 1,87 | | 1,76 | |
| 2,20 | | 2,29 | | 2,55 | | 2,17 | |
| 1,25 | yes | 3,60 | | 1,63 | | 1,55 | |
| 2,48 | | 3,03 | | 1,84 | | 2,06 | |
| 1,12 | yes | 2,12 | | 1,31 | Yes | 1,77 | |
| 1,75 | | 2,30 | | 0,79 | Yes | 1,62 | |
| 1,67 | | 1,36 | Yes | 1,55 | | | |
| 1,43 | | 2,91 | | 1,98 | | | |
| 2,41 | | 1,60 | | 1,89 | | | |
| 2,64 | | 2,82 | | 1,65 | | | |
| 2,49 | | 1,55 | | 1,78 | | | |
| 1,60 | Yes | 3,00 | | 1,21 | Yes | | |
| 2,95 | | 1,88 | | 1,86 | Yes | | |
| 2,67 | | 2,32 | | 1,07 | Yes | | |
| 2,31 | | 3,69 | | 2,24 | | | |
| 2,24 | | 2,93 | | 2,20 | | | |
| 2,45 | | 1,87 | | 1,08 | Yes | | |
| 1,49 | | 2,55 | | 1,46 | | | |
| 2,06 | | 1,70 | | 1,72 | | | |
| 3,67 | | 2,80 | | 2,29 | | | |
| 2,57 | | 3,74 | | 1,74 | | | |
| 2,33 | | 2,27 | | 0,92 | Yes | | |
| 1,50 | | 1,92 | | | | | |

*Table 1 continued on next page*

*Table 1 continued*

| H2O Intensity ratio | Pros + | *Btk*^ΔCry Intensity ratio | Pros + | *Btk*^SA11 Intensity ratio | Pros + | Crystals Intensity ratio | Pros + |
|---|---|---|---|---|---|---|---|
| 2,10 | | 1,79 | | | | | |
| 1,47 | | 2,39 | | | | | |
| 1,76 | | | | | | | |

processing of the Cry1A toxin family in the fly midgut fed with *Btk*^SA11 spores. We used an anti-Cry1A antibody raised against the activated form of the toxin, which therefore recognizes both forms (*Babin et al., 2020*). As a control, we incubated *Btk*^SA11 spores in vitro in water, at 25 °C for 4 hr, 1 days, 3 days and 6 days. Under these conditions, the protoxin form of Cry1A at 130kD was predominant and stable for at least 3 days before fading (control in *Figure 6D*, right part of the blot). This observation is in agreement with the fact that the half-life of Cry1A crystals has been estimated at about 1 week in soil or under laboratory conditions at 25 °C (*Hung et al., 2016*). We used the same initial *Btk*^SA11 spore preparation (0 hr) to feed the flies. We further dissected intestines and extracted total proteins at different times post-feeding. Interestingly, as early as 4 hr, we observed the 67kD activated form of the Cry1A toxins (*Figure 6D* left part). Noteworthy, the 130kD protoxin forms were still present, as the flies kept ingesting spores and crystals throughout the experiments. Six days after ingestion, almost no more protoxins or toxins were detectable due to the instability of the crystals (*Figure 6D*). Thus, our data show that the crystals can be processed in the midgut of adult *Drosophila* to give rise to active forms of the Cry1A toxins.

As mentioned previously, the increase in EEPs/EEs number was prominent in the posterior midgut (*Figure 2—figure supplement 1*). Indeed, it has been previously observed that posterior midgut is more prone to ISC proliferation and EE differentiation or tumor formation (*Beebe et al., 2010*; *Marianes and Spradling, 2013*; *Tamamouna et al., 2020*). Nonetheless, a differential processing of Cry1A protoxins along the midgut could also participate to this difference. Thereby, we fed flies with purified *Btk*^SA11 crystals and 24 hr later, we dissected and crushed the intestines by separating the anterior midgut from the posterior part. Furthermore, we performed subcellular fractioning by separating the soluble fraction (considered as the cytosol-enriched fraction) and the insoluble fraction containing membranes. As expected, actin is mainly found in insoluble fraction (lysis buffer detergent-free) and was consequently used as house-keeping protein for this cellular compartment. Interestingly, we observed that protoxins were already activated in the anterior part of the midgut (the 67kD form) but were only found in insoluble fraction, containing membranes (*Figure 6E*). In the posterior midgut, we detected a stronger quantity of the 67kD-activated form both in the soluble and insoluble fraction, suggesting an internalization by epithelial cells (*Figure 6E*). In this part of the midgut, smaller forms appeared mainly associated with the membrane, probably resulting from degradation processes. Notably, we did not detect the 130kD protoxins in the cytoplasmic fraction of both anterior and posterior midguts, suggesting that the protoxins remained associated with membranes. Moreover, total processing was never achieved since some protoxins still remained detectable. Altogether, our data show that crystals containing Cry1A protoxins are processed all along the adult *Drosophila* midgut to generate activated Cry1A toxins.

## Cry1A toxins cross the intestinal barrier independently of cell adhesion strength and cell death

DE-Cad/adherens junctions are located basally in the posterior *Drosophila* midgut (*Chen et al., 2018*; *Chen and St Johnston, 2022*; *Loudhaief et al., 2017*; *St Johnston and Ahringer, 2010*) and are therefore protected from direct contact with the luminal content (*i.e.* from the Cry1A toxins ingested along with the food) by the septate junctions (i.e. the tight junctions) located apically. To reach the ISC-progenitor pairs that are also located basally within the intestinal epithelium, Cry1A toxins can follow different routes. Indeed, toxins could weaken the septate junctions, penetrating the intercellular space to directly interact with DE-Cadherins. An alternative might be that Cry-induced cell death would cause a leaky midgut favoring the toxin passage. To address this question, we first overexpressed either DE-Cad or Connectin in ECs using *myo1A*^ts-*Gal4 UAS-GFP* driver. We assumed that

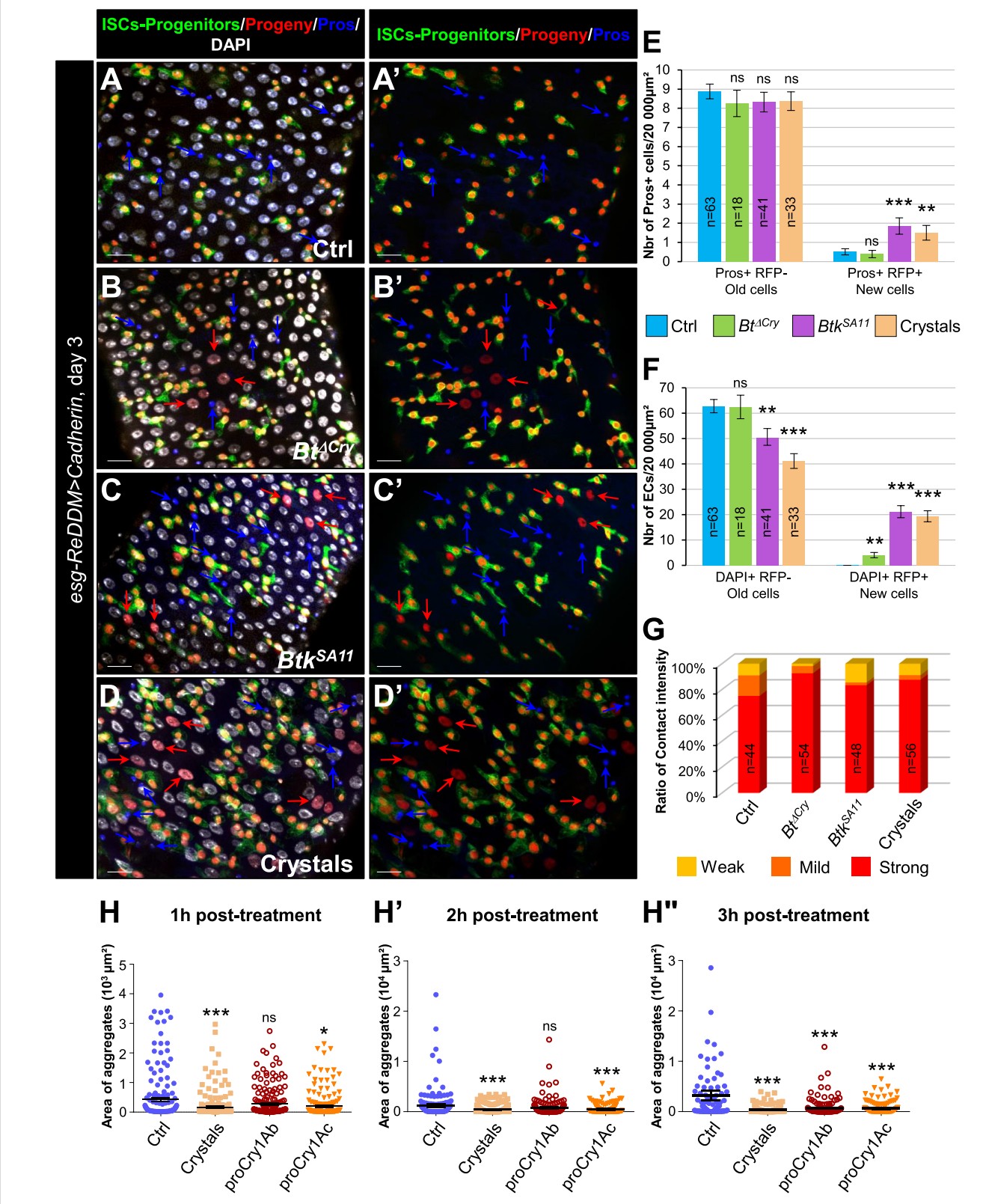

**Figure 5.** Increasing adherens junction strength rescues crystal-dependent cell fate diversion. (**A–G**) *esg-ReDDM >DE Cad Drosophila* midgut R4 region. These flies specifically overexpress the DE-Cad in ISCs and progenitors. Flies were fed with water (A-A', and blue in E and F, Ctrl), *Btk^{ΔCry}* spores (B, B', and green in E and F), *Btk^{SA11}* spores (C, C' and purple in E and F) or Crystal (D-D' and beige in E and F) and observed 72 h PI (see *Figure 3A* for the experimental design). In (**A-D'**) blue arrows point to old EEs and red arrows newborn ECs. (**E**) Number of old EEs (Pros + RFP-) and new EEs (Pros +

*Figure 5 continued*

RFP + ) and (**F**) number of old ECs (DAPI +RFP-) and new ECs (DAPI + RFP + ) in the conditions described in (**A–D**). (**A-D'**) 40 X magnification. Scale bar = 20 μm. (**G**) Graph representing the percentage of the different categories of cell contact intensity between ISCs and progenitors in the experimental conditions shown in *Figure 5—figure supplement 1A–D'*. Weak = Contact Intensity/Membrane Intensity <1.4; Mild = 1.4 < Contact Intensity/ Membrane Intensity <1.6; Strong = Contact Intensity/Membrane Intensity >1.6. n=number of cell pairs analysed. (**H-H"**) Cell aggregation assays on S2 cells expressing the DE-Cadherin::GFP. Cells placed under constant rotation were incubated with or without *Bt* crystals or purified Cry protoxins (Cry1Ab or Cry1Ac) for 1 hr (**G**), 2 hr (**G'**) or 3 hr (**G"**). Each scatter plot represents the area (μm2) of all objects (aggregates or individual cells) obtained from three independent experiments. Representative images of cell aggregates formed in aggregation assays are shown in *Figure 5—figure supplement 1F* data. In (**E and F**), n=number of 40 X images analyzed. In (**G**), n=number of cell pairs analyzed. Data is reported as mean ± SEM. ns (non-significant), * (p≤0.05), ** (p≤0.01), *** (p≤0.001).

The online version of this article includes the following source data and figure supplement(s) for figure 5:

**Source data 1.** Assessment of ISC division.

**Figure supplement 1.** Cry1A protoxins reduced homophilic interactions of DE-cadherin.

**Figure supplement 1—source data 1.** Assessment of ISC division.

**Figure supplement 2.** Connectin overexpression does not rescue cell adhesion disturbance induced by *Btk* crystals of toxins.

**Figure supplement 2—source data 1.** Counting of cell types.

increasing the amount of cell-cell adhesion molecules would strengthen the sealing of the intestinal epithelium, thereby limiting intercellular passage of Cry toxins. As expected, DE-Cad overexpression in ECs partially rescued the increased number of EEs due to *Btk*$^{SA11}$ or crystal ingestion (***Figure 7B–G and N*** and ***Figure 7—figure supplement 1A–F***) while, surprisingly, Connectin overexpression did not (***Figure 7H–J and N*** and ***Figure 7—figure supplement 1G–I***). These data suggest that strengthening EC-EC interaction on its own is not sufficient to prevent Cry toxins to reach basal ISC-progenitor cell junctions. However, the partial rescue obtained with the DE-Cad overexpression suggest that DE-Cad in ECs could trap Cry1A toxins away from the ISC-progenitor cell junctions, limiting the amount of toxins able to interfere. We then blocked EC death by overexpressing the anti-apoptotic factor p35 in ECs, but we did not rescue EE increase induced by *Btk*$^{SA11}$ or crystal ingestion (***Figure 7K–N*** and ***Figure 7—figure supplement 1J–L***). Together our data suggest that Cry1A toxins can reach the basally located ISC through routes that do not depend on cell adhesion strength or cell death. Further investigations are therefore needed to clearly identify the cellular mechanisms involved.

## Discussion

Our results show that the Cry1A toxin family of *Btk* disrupts the gut cellular homeostasis of the non-susceptible organism *Drosophila melanogaster*. Cry1A induces EC death coupled to an increase in ISC proliferation to replace the damaged ECs. Importantly, Cry1A toxins also altered intestinal cell composition by weakening DE-Cadherin-dependant cell-cell adhesion which is normally highly enriched in adherens junctions linking ISCs to their immediate progenitors (***Choi et al., 2011***; ***Maeda et al., 2008***; ***Ohlstein and Spradling, 2006***; ***Zhai et al., 2017***). As a consequence, progenitors are pushed toward the EE path of differentiation instead of the EC, owing to reduced activation of N signaling in progenitors (***Figure 8***; ***Guo and Ohlstein, 2015***; ***Maeda et al., 2008***; ***Ohlstein and Spradling, 2006***; ***Ohlstein and Spradling, 2007***; ***Pasco et al., 2015***; ***Zhai et al., 2017***). Our data confirmed that the duration and/or the strength of the cell-cell contact between the ISC and its progenitor daughter cell is important to drive the progeny toward an appropriate fate. Indeed, it has been shown both in *Drosophila* and in mammalian cell culture that adherens junctions are crucial to reinforce the contact between neighboring cells to allow the activation of N signaling (***Falo-Sanjuan and Bray, 2021***; ***Shaya et al., 2017***; ***Zhai et al., 2017***), N being necessary for the EB-EC cell fate choice.

Only the Cry1A family of toxins induces an increased number of EEs while Cry2A toxins do not display any phenotype. Previous studies showed that Cry1A and Cry2A bind to different receptors in the intestinal epithelium of susceptible lepidopteran larvae, and proteins of the Cadherin family appear essential for Cry1A, but not Cry2A, binding (***Adang et al., 2014***; ***Gao et al., 2019***; ***Hernández-Rodríguez et al., 2013***; ***Li et al., 2020***; ***Liu et al., 2018a***). *Drosophila* possesses 17 genes encoding for members of the Cadherin superfamily (***Hill et al., 2001***). Cad88C is the most similar to Bt-R Cadherin of susceptible Lepidoptera, but shares only 17% of identity, (***Stevens et al., 2017***) and is poorly expressed in the *Drosophila* midgut (see http://flygutseq.buchonlab.com/ and https://flygut.

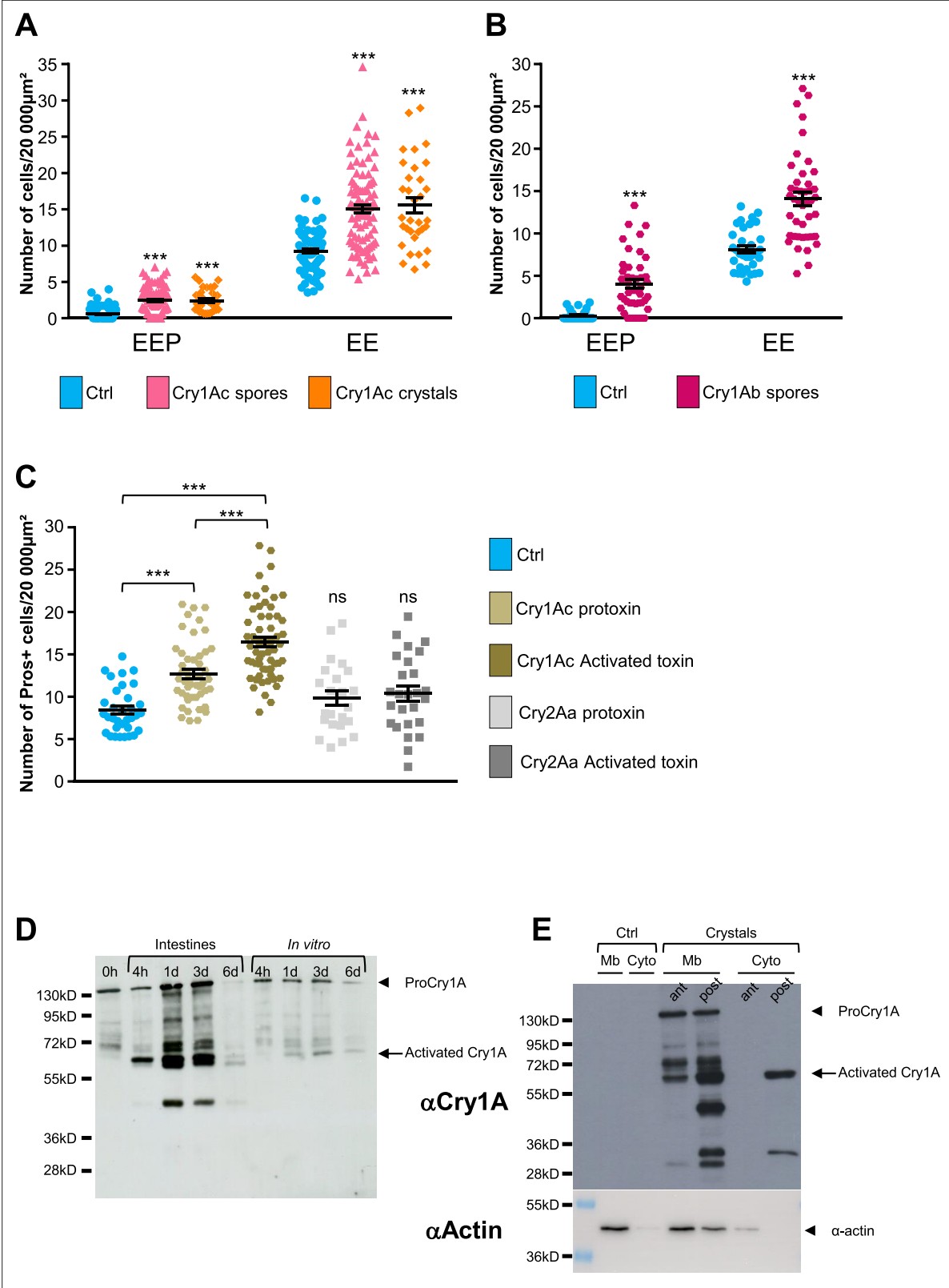

**Figure 6.** Cry1A toxins mimic *Btk* crystal effects. (**A–C**) *esg >GFP* flies fed with water (blue, Ctrl), *Btk*<sup>Cry1Ac</sup> spores (fuchsia in A), Cry1Ac crystals (orange in A), *Btk*<sup>Cry1Ab</sup> spores (rose in B), Cry1Ac protoxins (light khaki in C), Cry1Ac activated toxins (khaki in C), Cry2Aa protoxins (light grey in C) and Cry2Aa activated toxins (grey in C). ns (non-significant). Data is reported as mean ± SEM. *** (p≤0.001). (**A and B**) Density of EEPs or EEs in the R4 region 72 hr PI. (**C**) Density of Pros + cells in the R4 region 72 hr PI. (**D and E**) Western Blot from dissected intestines using a polyclonal Anti-Cry1A antibody

*Figure 6 continued on next page*

*Figure 6 continued*

detecting both the protoxins and the activated forms of Cry1A family of toxins.(**D**) (left lane) 0 h corresponds to *Btk*<sup>SA11</sup> spores extemporaneously resuspended in water. (Right part of the blot) *Btk*<sup>SA11</sup> spores incubated ex vivo (control) in water at 25 °C for the period of the experiment. We mainly detect the protoxin forms of Cry1A at 130 kDa (arrowhead). (Left part of the blot). Proteins extract from midguts of flies fed by the same *Btk*<sup>SA11</sup> preparation (T 0 h) at 4 hr and 1, 3, and 6 days PI. The 130 kDa protoxins are still visible. The 67 kDa activated form appears as early as 4 hr (arrow). 6 days PI no more toxins are detected in the midgut. (**E**) Flies fed 2 days with water (Ctrl, left part) or with purified crystals (right part). Protoxins (130 kDa) are present in the insoluble fraction (Mb) in both the anterior (ant) and the posterior (post) midgut. The 67 kDa activated forms are present in the insoluble fraction of both the anterior and posterior midgut and in the soluble fraction (Cyto) of the posterior midgut. Actin was used as western blot loading control, especially for the insoluble fraction.

The online version of this article includes the following source data and figure supplement(s) for figure 6:

**Source data 1.** Counting of cell types.

**Figure supplement 1.** Cry1A toxins mimic *Btk* crystal effects.

**Figure supplement 1—source data 1.** The uncropped images of Western blots.

epfl.ch/). However, Cry1A effects on *Drosophila* progenitor cell fate appear to depend specifically on the DE-Cadherin present in the adherens junctions, since overexpression of DE-Cadherin in progenitor cells (ISCs, EBs and EEPs) can overcome Cry1A-induced impacts while another component of the adherens junctions (Connectin) cannot. Interestingly, the percentage of identity between the Cry1A binding regions (CBRs) shared by the orthologs of Cadherin-type receptors in different susceptible lepidopteran species ranges from 21% to 66% (*Li et al., 2021*; *Shao et al., 2018*). Hence, the presence of a well-conserved primary consensus sequence within the CBRs cannot explain the specificity of binding. In agreement, it has been recently shown in susceptible Lepidoptera that only two dipeptides within the CBRs are essential for high-affinity binding of Cry1A to its Cadherin receptor (*Liu et al., 2018a*). These two dipeptides are not conserved in the CBRs of the Bt-R orthologs in different lepidopteran species targeted by Cry1A toxins (*Li et al., 2021*). These data suggest that binding of Cry1A to the receptor relies more on a conserved conformation of the CBRs than on a conserved primary sequence. In addition, alignment of the *Helicoverpa armigera* (a Cry1A susceptible lepidopteran) CBR sequence (*Gao et al., 2019*) and the DE-Cadherin sequence between amino acids 164 and 298 shows 24% identity and 47% similarity (using BlastP, https://blast.ncbi.nlm.nih.gov/). Altogether, these results support a possible binding of Cry1A toxins to *Drosophila* intestinal DE-cadherins, although this could occur with a low affinity.

Susceptibility of lepidopteran larvae to CryA1 toxins relies on the presence of a secondary receptor such as Alkaline phosphatases, Aminopeptidases N, or ABC transporters (*Adang et al., 2014*; *Gao et al., 2019*; *Li et al., 2020*; *Liu et al., 2018b*). None of the orthologs of these receptors has been shown to be strongly expressed in the *Drosophila* midgut (*Li et al., 2021*; *Stevens et al., 2017*), which could explain why binding of Cry1A to DE-cadherins does not lead to the death of adult *Drosophila*. Interestingly, in *Drosophila* larvae, heterologous expression of ABCC2 from *Bombix mori* or *Plutella xylostella*, or Aminopeptidases N from *Manduca sexta* was sufficient to induce Cry1A-dependent death (*Gill and Ellar, 2002*; *Obata et al., 2015*). Conversely, heterologous expression of the *Bombix mori* Cadherin Bt-R receptor in *Drosophila* is not sufficient to induce death upon exposure to Cry1A but strongly enhances death when co-expressed with BmABCC2 (*Obata et al., 2015*). Together, these data suggest that, in *Drosophila,* in the absence of a Cadherin receptor displaying a high affinity to Cry1A toxin, endogenous Cadherins with reduced affinity toward Cry1A could step in to enhance the death potential of Cry1A toxins in the context of heterologous expression of a secondary receptor from susceptible Lepidoptera. In agreement, it has been previously demonstrated in *Drosophila* larvae that increasing the dose of ingested *Btk* spores and crystals could ultimately lead to death (*Babin et al., 2020*; *Cossentine et al., 2016*), arguing that the receptors (i.e. the Cadherins and the secondary receptors) present in the *Drosophila* midgut have lower affinities for Cry1A toxins.

Cry toxin activities in susceptible organisms rely not only on the presence of specific host receptors in the midgut, but also on the extreme midgut pH and reducing environment allowing crystal solubilization, as well as the enzymatic capacity of digestive proteins, both of which are involved in the conversion of protoxins into active toxins (*Fiuza et al., 2017*; *Mendoza-Almanza et al., 2020*; *Pardo-López et al., 2013*; *Shao et al., 2013*). Nevertheless, our data along with another recent study (*Stevens et al., 2017*) suggest that crystals can be solubilized and then protoxins activated in the *Drosophila* midgut. Therefore, more investigations will be necessary to understand how crystals are

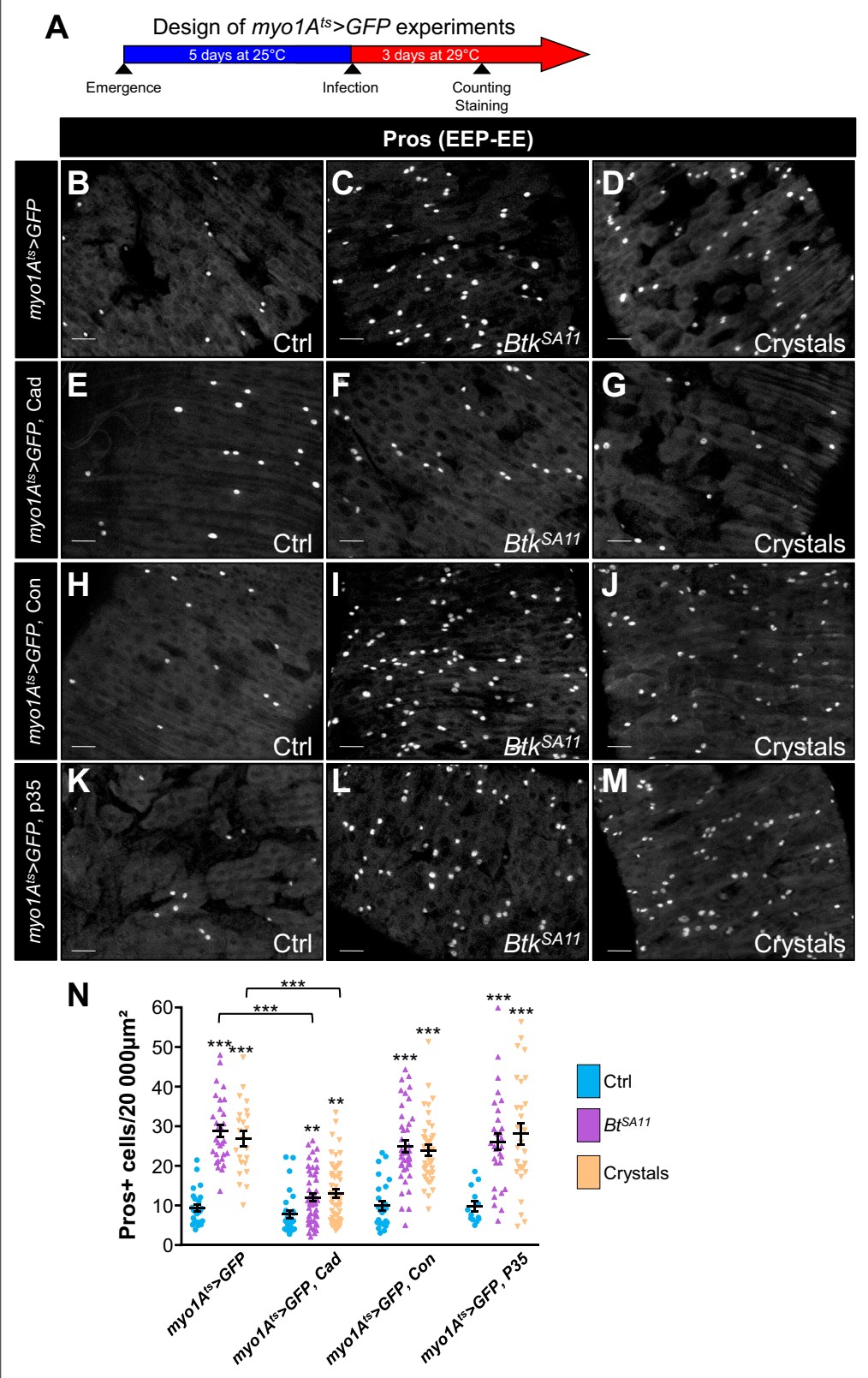

**Figure 7.** Cry1A toxins likely cross the intestinal barrier through EC transcytosis. (**A**) Schema of the experimental design for the *myo1A-GAL4 UAS-GFP tub-GAL80^ts* (*myo1At^s >GFP*) overexpression in ECs used in this entire figure. (**B–N**) R4 region of midguts of flies fed with water (Ctrl, B, E, H, K and blue in N), *Btk^SA11* spores (C, F, I, L and green in N) and crystals (D, G, J,M and beige in N) and labelled for Pros. ×40 magnification. Scale bar =

*Figure 7 continued on next page*

*Figure 7 continued*

20 µm. (**B–D**) *myo1At^s >GFP* midguts. (**E–G**) *myo1At^s >GFP* midguts overexpressing DE-Cad. (**H–J**) *myo1At^s >GFP* midguts overexpressing Connectin (Con). (**K–L**) *myo1At^s >GFP* midguts overexpressing the anti-apoptotic p35 factor. (**N**) Counting of EEPs/EEs (Pros + cells) in the different conditions described in (**B–M**). Data is reported as mean ± SEM. ** (p≤0.01), *** (p≤0.001).

The online version of this article includes the following source data and figure supplement(s) for figure 7:

**Source data 1.** Counting of Prospero positive cells.

**Figure supplement 1.** Cry1A toxins likely cross the intestinal barrier through EC transcytosis.

processed and protoxins activated in the intestine of non-susceptible organisms. What makes the difference between Cry-susceptible and non-susceptible organisms is likely the affinity of Cry toxins for midgut host receptors. The higher the affinity of the toxins, the greater the cellular damage/death and the greater the risk of death. In addition, the capacity of regeneration of the midgut epithelium also plays an important role to overcome *Bt* pathogenicity (***Castagnola and Jurat-Fuentes, 2016***). If the regeneration of the intestinal epithelium is more efficient than the destructive capacity of Cry toxins, the host will survive. However, whatever the host is (susceptible or non-susceptible), *Bt* uses a three-pronged strategy to improve its degree of virulence. First, it tries to disrupt the epithelial barrier function by damaging or killing ECs. Second, it diverts the behavior of the progenitor cells toward the wrong fate, thus limiting the amount of ECs produced that are necessary to replace the damaged ones and to maintain the midgut integrity. Third, killing ECs also reduces the capacity of these cells to produce reactive oxygen species and antimicrobial peptides known to be involved in antimicrobial defenses (***Allaire et al., 2018***; ***Capo et al., 2016***; ***Kim and Lee, 2014***).

The *Bt* subspecies *kurstaki* and *aizawai* are widely used as microbial pesticide to fight lepidopteran pests worldwide (***Casida and Bryant, 2017***), both subspecies partially producing the same Cry1A toxins (***Caballero et al., 2020***). Likewise, 180 millions of hectares of genetically modified crops express

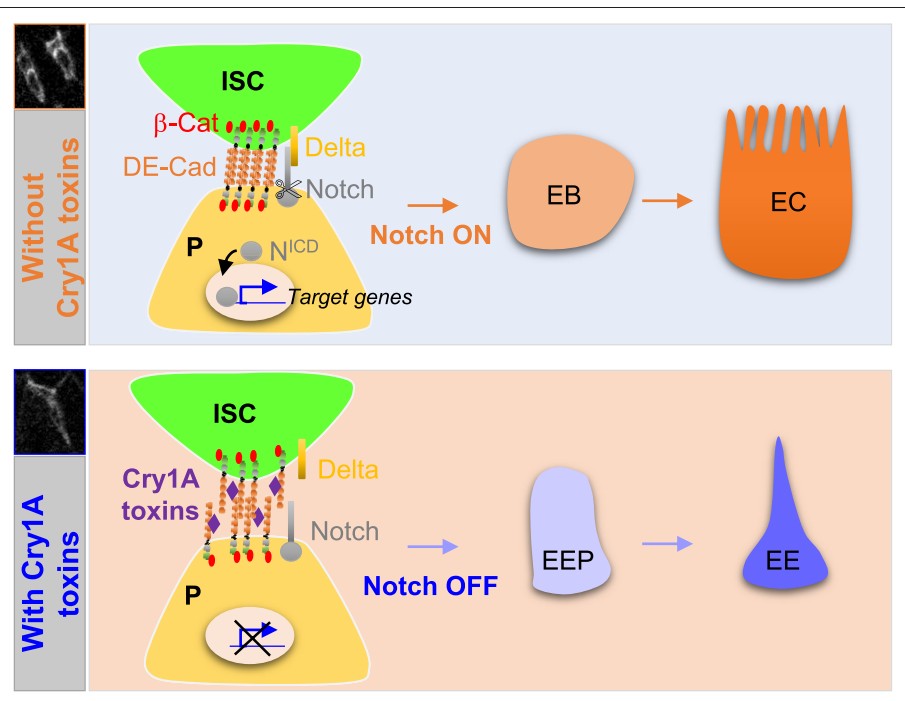

**Figure 8.** Cry1A toxins interfere with progenitor fate behavior. **Notch ON:** in *Drosophila*, 90% of ISC daughter cells commit to the EB/EC fate owing to the strong activation of the Notch signaling pathway in the EBs. The adherens junction DE-Cadherin (DE-Cad)-dependent are required to permit the interaction between the Delta ligand in ISC and the Notch receptor in EB. **Notch OFF:** Ingestion of Cry1A toxins impedes the DE-Cad homophilic interaction between the ISCs and their progenitor daughter cells, reducing the activation of Notch signaling in progenitors. Consequently, progenitors adopt an EEP/EE fate.

Cry1A toxins. Consequently, Cry1A toxins are present in the food, the feed and in the environment, implying that many organisms might be affected. As the mechanisms of intestinal progenitor fate choice are conserved in the animal kingdom (*Guo et al., 2021*; *Joly and Rousset, 2020*; *Zwick et al., 2019*), it would be interesting to investigate whether Cry1A toxins can also promote an increased number of EEs in other organisms (vertebrates and invertebrates). EEs, through the production of neuropeptides and hormones, are involved in the regulation of many physiological functions such as feeding behavior, metabolism and immune response (*Guo et al., 2021*; *Nässel and Zandawala, 2019*; *Watnick and Jugder, 2019*). Consequences of this increase in EE number could be, for example, metabolic dysfunctions or inflammatory pathologies. More studies are needed to understand the physiological impacts of this change in intestinal cellular composition on organismal health.

## Materials and methods

### Bacterial strains

*Btk$^{\Delta Cry}$* (identified under the code 4D22, *González et al., 1982*), *Btk$^{Cry1Ac}$* (identified under the code 4D4), and the *E. coli* strains producing Cry1Ac (identified under the code ECE53), Cry1Ab (identified under the code ECE54) and Cry2Aa (identified under the code ECE126) were obtained from the Bacillus Genetic Stock Center (https://bgsc.org/). The strain *Btk* SA-11 (*Btk$^{SA11}$*) was isolated from the commercial product Delfin.

### Generation of *Btk$^{Cry1Ab}$*

The mutant *Btk$^{Cry1Ab}$* producing only Cry1Ab as crystal toxin was obtained from the WT strain *Btk$^{SA11}$*, by a procedure of plasmid curing, as follows. After isolation on TSA-YE agar (Biomérieux, 18 hr culture at 30 °C), the strain *Btk$^{SA11}$* was sub-cultured successively 3 times in 10 mL of brain heart Infusion (BHI, Oxoid) broth at 42 °C with agitation, for 64, 48, and 36 hr, respectively. The first BHI culture was inoculated from an isolated colony, and the subsequent cultures were inoculated with 100 µl of the previous ones. Clones from the last culture were isolated on TSA-YE agar after serial dilution, then subcultured on the sporulating medium hydrolysate of casein tryptone (HCT) + 0.3% Glc, in order to verify the absence of crystal production, using phase-contrast microscopy (NF EN ISO 7932/Amd1). A panel of crystal-negative isolates were then subjected to whole genome sequencing as described hereafter. The genomic DNA of *Btk$^{Cry1Ab}$* and *Btk$^{SA11}$* was extracted using the KingFisher cell and Tissue DNA kit (ThermoFisher) and sequenced using Illumina technology at the Institut du Cerveau et de la Moelle Epinière (ICM) platform, as previously described (*Bonis et al., 2021*), (NCBI accession numbers SAMN23436140 and SAMN23455549, respectively). The mutant *Btk$^{Cry1Ab}$* was selected for the single presence of *cry1Ab* as *cry* toxin plasmid gene. The absence of *cry* genes in *Btk$^{Cry1Ab}$*, with the exception of *cry1Ab*, was confirmed from raw reads using KMA (*Clausen et al., 2018*).

### Spore production

From isolated colonies on LB agar Petri dish, 4x5 mL of *Bt* pre-culture was carried out. The pre-culture was used for sowing 4x500 mL of PGSM medium (0.75% casamino acids, 0.34% $KH_2PO_4$, 0.435% $K_2HPO_4$, 0.75% glucose, 1.25 mM $CaCl_2$, 0.123% $MgSO_4$, 0.002% $MnSO_4$, 0.014% $ZnSO_4$, 0.02% $FeSO_4$) and allowed to grow and sporulate in an incubator shaker at 30 °C, 180 rpm for 2 weeks. In order to eliminate vegetative cells, culture was heated 1 hr at 70 °C and then centrifuged 15 min at 7500 g. The pellet was resuspended with a Dounce homogenizer in 1 L of 150 mM NaCl and placed for 30 min on roller agitation at room temperature. After centrifugation (15 min, 7500 g), the pellet was washed twice with sterile water. The final pellet was resuspended in 30 mL of sterile water, dispatched in 1 mL weighed tubes and lyophilized for 24–48 hr. The spore mass was determined by the difference between the full and the empty tubes weights.

### Spore titer

The lyophilized spores were resuspended in sterile water to obtain a concentration of 50 mg/mL. This solution was diluted serially (100 µL in 900 µL of sterile water) to obtain $10^{-1}$ to $10^{-9}$ dilutions. 100 µL of dilutions $10^{-5}$ to $10^{-9}$ were plated on LB agar and incubated at 30 °C overnight. The number of colonies was counted for each dilution and reported to the mass of spores plated. The experience

was renewed three times. The mean of these ratios allows us to determine the titer of spores in CFU (colony forming unit)/g.

## Crystal purification

A total of 2x1 g of $Btk^{SA11}$ lyophilized spores were resuspended in 2x30 mL of sterile water and placed on roller agitation at 4 °C for 5 hr. Then, the solution was sonicated for 4 cycles of 15 s/15 s with a frequency of 50% (Fisherbrand Model 505 Sonic Dismembrator). 6x10 mL were deposited on a discontinuous sucrose gradient (67%/72%/79%) and centrifuged overnight in SW28 swinging buckets at 100,000 g at 4 °C (ultracentrifuge Thermo-Sorval WX Ultra 80). The crystals were collected at the 72%/79% and 72% /67% interfaces with a micropipette and dispatched by 10 mL in centrifuge tubes (Beckman Avanti JE, rotor JA 25.50). A total of 25 mL of sterile water were added in each tube, vortexed and centrifuged at 4 °C at 16,000 g for 15 min. Each pellet was resuspended with 20 mL of sterile water and centrifuged again in the same conditions. Each final pellet was resuspended in 2 mL of sterile water, aliquoted by 1 mL in weighed microtubes and lyophilized 24 hr-48hr. The crystal mass was determined by the difference between the full and the empty tubes' weights.

## Cry protoxin production

2x15 mL of LB ampicillin (50 mg/mL) were inoculated with two colonies of *E. coli* expressing the desired Cry toxin, and allowed to grow at 37 °C, 220 rpm overnight. 4x5 mL of the overnight preculture were added to 4x500 mL of LB ampicillin (50 mg/mL) and allowed to grow at 37 °C and 200 rpm until $DO_{600}$=0.6–0.7. Cry protein expression was induced by adding 500 µL of 1 M IPTG (isopropyl β-D-1-thiogalactopyranoside) in each culture. The cultures were left at 37 °C and 200 rpm overnight and then centrifuged at 6500 g, 15 min at 22 °C. Pellets were pooled in two batches and resuspended 100 mL of cold WASH buffer (20 mM Tris, 10 mM EDTA, 1% Triton X-100, pH 7.5) and incubated 5 min at 4 °C. 500 µL of lysozyme (20 mg/mL) were added in each solution and incubated 15 min at 30 °C and then 5 min at 4 °C. The solutions were sonicated for 6 cycles of 15 s/15 s at 40% (Fisherbrand Model 505 Sonic Dismembrator) and centrifuged 10 min at 10,000 g, at 4 °C. The pellets were washed twice with 100 ml of WASH buffer and centrifuged in the same conditions. The last pellets were weighed, resuspended in CAPS buffer (50 mM CAPS, 0.3% lauroyl-sarcosine, 1 mM DTT pH11) to obtain a final concentration of 100 mg/mL, placed for 30 min under roller agitation at room temperature and centrifuged 10 min at 10,000 g, 4 °C. The supernatant was dialyzed twice against 50 volume of PBS1x, 0.1 mM DTT and twice against 50 volume of PBS1X at 4 °C and centrifuged 10 min at 20,000 g at 4 °C. The supernatant was conserved at –20 °C until purification or digestion (activation) by trypsin.

## Cry toxin activation

Half of the produced supernatant (see above) was dosed by the Bradford method (Protein Assay Dye Reagent Concentrate, Biorad #500–0006) and digested with 1% trypsin (weight/weight) (trypsin from bovine pancreas, Sigma #T1005) at 37 °C for 72 hr. The Cry toxins were then purified by FPLC (after 72 hr, the trypsin is fully degraded).

## Cry toxin purification

The Cry toxins produced from *E. coli* (activated or not) were purified by FPLC (Äkta, UPC900/P920/INV907/FRAC950) on a 1 mL benzamidine column (HiTrap Benzamidine FF (high sub), GE Healthcare #17-5143-01) with PBS1X as charge buffer, PBS1X, 1 M NaCl as buffer for non-specific link and 100 mM glycin pH 3 as elution buffer.

## Fly stocks and genetics

The following stocks are listed at the Bloomington *Drosophila* Stock Center (https://bdsc.indiana.edu/): WT canton S (#64349). *w; Sco/CyO; tub-GAL80ts/TM6b* (#7018). *w; tub-GAL80ts; TM2/TM6b* (#7019). *w; esg-GAL4NP5130* (#67054). *w; UAS-GFP/TM3 Sb* (#5430). *w; UAS-shg-R* (DE-Cadherin) (#58494); *y w, shg::Tomato* (#58789).

### Other stocks

*w;; Dl-GAL4/TM6b* (**Zeng et al., 2010**). *w; tub-GAL80ts; Dl-GAL4 UAS-GFP/TM6b* (this study). *w; esg-GAL4NP5130 UAS-GFP* (**Shaw et al., 2010**). *w; esg-GAL4NP5130 UAS-GFP; tubGAL80ts* (**Apidianakis**

*et al., 2009*). *w; Su(H)GBE-GAL4, UAS-CD8::GFP* (*Zeng et al., 2010*). *w; Su(H)GBE-GAL4/SM6β; tub-GAL80ᵗˢ UAS-GFP/TM6b* (this study). *w; myo1A-GAL4* and *w; myo1A-Gal4; tubGal80ts UAS-GFP/TM6b* (*Shaw et al., 2010*). *w; myo1A-GAL4 UAS-GFP/CyO* (*Apidianakis et al., 2009*). *w; UAS-GFP::CD8; UAS-H2B::RFP/TM2* (*Antonello et al., 2015*). *w; UAS-CD8::GFP; UAS-H2B::RFP, tub-GAL80ᵗˢ/TM2* (*Antonello et al., 2015*). *w; esg-GAL4, UAS-CD8::GFP; UAS-H2B::RFP, tub-GAL80ᵗˢ/TM6b* (*Antonello et al., 2015*). *w; UAS-CD8::GFP; Dl-GAL4, UAS-H2B::RFP/TM6b* (this study). *w;; UAS-GC3Ai^{G7S}* (*UAS-Casp::GFP*) (*Schott et al., 2017*). *w; UAS-connectin* (*Zhai et al., 2017*). *UAS-p35* (*Amcheslavsky et al., 2009*).

## Cell lineage

### *Dl-ReDDM* experiments
*w; tub-GAL80ᵗˢ; Dl-GAL4 UAS-GFP/TM6b* females were crossed to *w; UAS-GFP::CD8; UAS-H2B::RFP/TM6* males at 18 °C. Progeny were kept at 18 °C until emergence. Flies were transferred at 25 °C for 5 days before infection, and then transferred at 29 °C for 2 days (*Figure 1—figure supplement 1C*).

### *esg-ReDDM* experiments
*w; esg-GAL4 UAS-GFP; UAS-H2B::RFP, tub-GAL80ᵗˢ/TM6b* females were crossed to WT males at 18 °C. Progeny were kept at 18 °C until emergence. Flies were transferred at 25 °C for 5 days before infection, and then transferred at 29 °C for 3 days (*Figure 3*).

### *Su(H)-ReDDM* experiments
*w; Su(H)-GAL4/SM6β; tub-GAL80ᵗˢ UAS-GFP/TM6b* females were crossed to *w; UAS-GFP::CD8; UAS-H2B::RFP/TM6* males at 18 °C. Progeny were kept at 18 °C until emergence. Flies were transferred at 25 °C for 5 days before infection, and then transferred at 29 °C for 3 days (Figure S3).

### DE-Cadherin and Connectin overexpression
*w; esg-Gal4 UAS-GFP; UAS-H2B::RFP, tub-GAL80ᵗˢ/TM6b* females were crossed to *UAS-DE-Cadherin* males. Progeny were kept at 18 °C until emergence. Flies were transferred at 25 °C for 5 days before infection, and then transferred at 29 °C for 3 days (*Figure 5 Figure 5—figure supplement 1* and *Figure 5—figure supplement 2*).

## *Drosophila* rearing and oral infection
*Drosophila* were reared on standard medium (0.8% Agar, 2.5% sugar, 8% corn flour, 2% yeast) at 25 °C with a 12 hr light/12 hr dark cycle. For oral infection, after 2 hr of starvation to synchronize the food intake, 5- to 6-day-old non-virgin females were transferred onto a fly medium vial covered with a filter disk soaked with water (control) or a suspension of spores (corresponding to $10^6$ CFU of spores per 4 cm² and per individual female; *Loudhaief et al., 2017*; *Nawrot-Esposito et al., 2020*), crystals, protoxins, or activated toxins. The quantity of crystals, protoxin, or activated toxins deposited on the filter disc corresponded to 30% of the spore weight, *Btk* crystals representing between 25% and 30% of the total weight of the 1:1 spore/crystal mix (*Agaisse and Lereclus, 1995 Monro, 1961*; *Murty et al., 1994*). Flies were kept feeding on the contaminated media until dissection in all the experiments.

## Dissection, immunostaining and microscopy
Dissection, fixation and immunostaining were performed as described by *Micchelli, 2014*. Dilutions of the various antibodies were: mouse anti-Armadillo N27A1 at 1:50 (DSHB), mouse anti-Connectin-C1-427 at 1/200 (DSHB), mouse anti-Prospero MR1A at 1:200 (DSHB), rabbit anti-PH3 at 1:1000 (Millipore, 06–570), Rabbit anti-Cleaved Caspase-3 at 1/600 (Cell Signalling Asp175 #9661), Goat anti-mouse AlexaFluor-647 at 1/500 (Molecular Probes Cat# A-21235), Goat anti-mouse AlexaFluor-546 at 1/500 (Molecular Probes Cat# A-11003), Goat anti-rabbit AlexaFluor-647 at 1/500 (Thermo Fisher Scientific Cat# A32733), Goat anti-rabbit AlexaFluor-546 at 1/500 (Thermo Fisher Scientific Cat# A-11010). For microscopy, guts were mounted in Fluoroshield DAPI medium (Sigma, # F6057) and immediately observed on a Zeiss Axioplan Z1 with Apotome 2 microscope. Images were analyzed

using ZEN (Zeiss), ImageJ and Photoshop software. Image acquisition was performed at the microscopy platform of the Sophia Agrobiotech Institute (INRAE1355-UCA-CNRS7254 – Sophia Antipolis).

## DNA constructs

The full-length expression construct of DE-cadherin Full Length fused to GFP (DEFL) was introduced into pUAST as previously described (*Oda and Tsukita, 1999*).

## Cell aggregation assay

*Drosophila* Schneider S2 cells (S2-DRSC, DGRC Stock 181; https://dgrc.bio.indiana.edu//stock/181) were obtained from DGRC (*Drosophila* Genomics Resource). We tested this cell line for contamination of mycoplasma (MycoFluor Mycoplasma Detection Kit, Invitrogen). We have not tested cell line identity because (1) S2 is the only cell line used in our lab and in a dedicated culture cell room, (2) this is the only *Drosophila* cell line present at the institute, (3) all experiments were conducted using the same batch of initial frozen ampoule received from the DGRC, and (4) transfection results were compared with control data experiment conducted in the same culture passage at the same time. S2 cells were cultured in Schneider's medium supplemented with 10% heat-inactivated fetal bovine serum (FBS) at 25 °C in a non-humidified ambient air-regulated incubator (*Gallet et al., 2006*). For S2 aggregation assay, $2.2 \ 10^6$ cells were plated in 25 cm² flask for each condition. After 6 hr, transient transfection was performed by mixing transfection reagent (TransIT–2020; Mirus Bio) with a reagent-to-DNA ratio of 3:1. A total of 3 µg plasmid DNA per T25 was used, corresponding to a 5:1 mixture of pUAST-DEFL and pWA-GAL4. Approximately 46 hr after transfection, the cells were collected into 15 mL tubes and centrifuged for 5 min at 400 g. The pellet was resuspended in 2 mL of fresh medium supplemented in $CaCl_2$ to obtain a final concentration of 7.4 mM and separated into single cells by repeated pipetting. Of this cell suspension, 500 µL were added to a well of a 24-well microplate. To allow cell-cell adhesion (aggregation), the microplate was placed under constant agitation on a rotary platform at 150 rpm at 25 °C for the indicated time (1 hr, 2 hr, and 3 h) with or without protoxins (at a final concentration 35 µg/mL). Cell aggregates formed in the wells were observed using an inverted Fluorescent microscope (Nikon, Eclipse TE2000-U). Images for the florescence of GFP was acquired using a CCD camera (ORCA ER, Hamamatsu Photonics). The same parameter settings were used to acquire images (objectives, gain, exposure time …). The area of fluorescent aggregates and individual cells were measured using Fiji software (*Schindelin et al., 2012*). The average area of a S2 cell in our condition is about 15.5 µm². Hence, quantification of the aggregates area was performed excluding all objects smaller than 15 µm². The mean area of aggregates were calculated after background subtraction. Three independent experiments were performed for each condition. Values in µm² were represented in GraphPad Prism 7 software as scatter-plot view. Statistical analysis was conducted using GraphPad Prism 7 software. The significance of the difference between CTR and exposed conditions was assessed using one-way ANOVA and Tukey's post hoc tests. Statistical parameters for each experiment can be found within the corresponding figure legends.

## Western blot

*Figure 6D* Twenty *Drosophila* (5–6 day-old non-virgin females) were orally infected with spores and reared at 25 °C with a 12 hr/12 hr day/night cycle for the indicated time. At the same time, $10^7$ CFU of spores in 50 µL of sterile water were incubated in the same conditions. After 4, 24, 72, or 144 hr, *Drosophila* midguts were dissected in PBS1x with anti-proteases (cOmplete Tablets EDTA free EASY Pack, Roche #04693132001). Then midguts were transferred on ice into a 2 ml microtube containing 200 µL of PBS1X with anti-proteases and crushed one minute at 30 Hz with a Tissue Lyser (Qiagen, Tissue Lyser LT).

*Figure 6E* Twenty *Drosophila* (5–6 day-old non-virgin females) were orally infected with water or purified crystals and reared at 25 °C with a 12 hr/12 hr day/night cycle for the indicated time. Flies were dissected and anterior and posterior regions of midguts were separated and lysed in a hypotonic buffer (25 mM HEPES, pH7.5, 5 mM $MgCl_2$, 5 mM EDTA, 5 mM DTT, 2 mM PMSF, 10 µg/mL leupeptin, 10 µg/mL pepstatin A) on ice. Midguts were crushed one minute at 30 Hz with the Tissue Lyser. Homogenates were centrifuged at 20,000 g for 45 min at 4 °C. The supernatant was the soluble fraction (considered as cytosol-enriched fraction) and the pellet was the insoluble fraction (containing membranes).

Proteins were dosed by Bradford method (Protein Assay Dye Reagent Concentrate, Biorad #500–0006). Twenty µg of midgut proteins and $2.10^4$ CFU of spores were deposited on a 8.5% SDS-polyacrylamide gel. After migration at 100 V for 1h30, proteins were transferred onto a PVDF membrane (Immobilon-P Membrane, Millipore #IPVH00010) (120mA/gel, 1 hr) in a semi-dried transfer system with a transfer buffer (200 mM glycine, 25 mM Tris base, 0.1% SDS, pH7.4, 20% methanol). Membranes were saturated with 5% milk in TBS-T (140 mM NaCl, 10 mM Tris base, 0.1% Tween 20, pH 7.4) for 1 hr and incubated overnight at 4 °C with a homemade anti-Cry1A antibody (*Babin et al., 2020*) or anti-actin monoclonal antibody (ACTN05, C4, Thermo Fisher Scientific) diluted at 1/7500 in TBS-T 3% BSA. After three washes of 10 min each with TBS-T, membranes were incubated with an anti-rabbit antibody coupled with HRP (Goat Anti-Rabbit (IgG), Invitrogen #G21234) diluted 1/7500 in TBS-T 2% milk for 1 hr at room temperature. Membranes were rinsed three times with TBS-T and once with TBS. Western blots were revealed with enhanced chemiluminescence (luminol and hydrogen peroxide, homemade) on an autoradiographic film (Amersham Hyperfilm ECL, GE Health-care #28906837).

## Measurement, counting, and statistical analysis

In all the data presented, the pictures and counting were always performed in the posterior part of the R4 region (http://flygut.epfl.ch/) named R4bc in the flygut site (see also *Buchon et al., 2013*; *Marianes and Spradling, 2013*; *Figure 1—figure supplement 1A*). Experiments were independently repeated at least three times.

## Cell type counting

midgut images were taken at ×40 magnification within the R4bc region with same microscope settings. A region of interest (ROI) of about 20,000 µm² were applied in which all cells were DAPI labelled. We utilized the Zeiss ZEN 2 (blue edition) to quantify cell number positive for a given marker. Data were represented as cell density (e.g. number of cells/20,000 µm²) except in *Figure 1—figure supplement 2* where ratio of each cell type was represented over total cell number. For a given cell type, when we noticed changes (increase or decrease) in both cell density and ratio, we assumed that this corresponded to global changes in the so-called cell type number.

## Cell junction intensity measurement

For both Arm or Tomato::DE-Cad labelling, we compared the average staining intensity at the junction between neighboring progenitors with the average intensity around the rest of cell membrane and then calculate the ratio between both values (see *Figure 4—figure supplement 1F, G*). Only junctions between individual paired cells were analyzed (we excluded clusters containing more than two cells). To assess junction strengths, we fixed the following scale: a weak junction when the ratio of staining intensity was <1.4, a mild junction when the ratio was between 1.4 and 1.6 and a strong junction when the ratio was >1.6.

## Statistics

Effects of treatments were analyzed using Kyplot or GraphPad Prism 7. When 'n' was equal or superior to 30, statistical analysis was performed using a parametric t-test. An F-test was systematically done before applying the t-test to verify the homogeneity of variances. When 'n' was inferior to 30, we used the non-parametric pairwise comparisons of the Mann-Whitney test. Differences were considered significant when *$p \leq 0.05$, **$p \leq 0.01$, ***$p \leq 0.001$. Error bars in all the graphics correspond to standard error of the mean (SEM).

# Acknowledgements

We are grateful to all members of the BES team for fruitful discussions. We want to thank Arnaud Felten (GVB Unit, Anses Ploufragan) and Pierre-Emmanuel Douarre (SEL unit, Anses Maisons-Alfort) for their help regarding the WGS analysis of the mutant *Btk*[Cry1Ab]. We also thank Laurent Ruel for providing pWA-Gal4. We are grateful to Hiroki Oda for providing the pUAST-DEFL. We thank Olivier Pierre from the microscopy platform of the Sophia Agrobiotech Institute for his help. Our thanks to the Université Côte d'Azur Office of International Scientific Visibility for English language editing of the manuscript. We also thank the Space, Environment, Risk and Resilience Academy of Université

Côte d'Azur for their financial support. This work has been supported by the French government, through the UCAJEDI Investments in the Future project managed by the National Research Agency (ANR) with the reference number ANR-15-IDEX-01 and through the ANR-13-CESA-0003–01, by the Région Provence Alpes Côte d'Azur, by the Département des Alpes-Maritimes, by the Institut Olga Triballat (PR2016-19) and by the ANSES PNR-EST & ECOPHYTO II (EST-2017–2021). RJ was funded by the association AZM & SAADE (Lebanon) and Université Côte d'Azur (ATER). RL was funded by the Ministère de l'Education Nationale, de l'Enseignement Superieur et de la Recherche (MESR) and a grant from the Fondation pour la Recherche Médicale (FRM).

## Additional information

### Funding

| Funder | Grant reference number | Author |
| --- | --- | --- |
| Agence Nationale de la Recherche | ANR-15-IDEX-01 | Armel Gallet |
| Agence Nationale de la Recherche | ANR-13-CESA-0003-01 | Armel Gallet |
| Conseil Régional Provence-Alpes-Côte d'Azur | | Armel Gallet |
| Conseil Départemental des Alpes Maritimes | | Armel Gallet |
| Institut Olga Triballat | | Armel Gallet |
| Office Français de la Biodiversité | | Armel Gallet |
| Azm and Saade Association | | Dani Osman |
| Université Côte d'Azur | | Rouba Jneid |
| Ministère de l'Education Nationale, de l'Enseignement Superieur et de la Recherche | | Rihab Loudhaief |
| Fondation pour la Recherche Médicale | | Rihab Loudhaief |

The funders had no role in study design, data collection and interpretation, or the decision to submit the work for publication.

### Author contributions

Rouba Jneid, Conceptualization, Data curation, Formal analysis, Investigation, Visualization, Writing – original draft; Rihab Loudhaief, Conceptualization, Investigation, Visualization; Nathalie Zucchini-Pascal, Formal analysis, Investigation, Visualization; Marie-Paule Nawrot-Esposito, Methodology; Arnaud Fichant, Data curation, Formal analysis; Raphael Rousset, Funding acquisition, Validation, Writing – review and editing; Mathilde Bonis, Resources, Data curation, Methodology; Dani Osman, Conceptualization, Funding acquisition, Validation, Writing – review and editing; Armel Gallet, Conceptualization, Data curation, Formal analysis, Supervision, Funding acquisition, Validation, Visualization, Project administration, Writing – review and editing

### Author ORCIDs

Armel Gallet ⓘ http://orcid.org/0000-0002-2054-4780

### Decision letter and Author response

Decision letter https://doi.org/10.7554/eLife.80179.sa1
Author response https://doi.org/10.7554/eLife.80179.sa2

## Additional files

### Supplementary files
- MDAR checklist

### Data availability
Genomic DNA of BtkCry1Ab and BtkSA11 was sequenced and is available at NCBI accession SRR17044437 and SRR17036893, respectively. All data generated or analysed during this study are included in the manuscript and supporting files.

The following datasets were generated:

| Author(s) | Year | Dataset title | Dataset URL | Database and Identifier |
|---|---|---|---|---|
| Fichant A, Bonis M | 2023 | SA11 | https://www.ncbi.nlm.nih.gov/sra/?term=%09SRR17044437 | NCBI Sequence Read Archive, SRR17044437 |
| Fichant A, Bonis M | 2023 | SA11cry1Ab | https://www.ncbi.nlm.nih.gov/sra/?term=SRR17036893 | NCBI Sequence Read Archive, SRR17036893 |

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

# Appendix 1

### Appendix 1—key resources table

| Reagent type (species) or resource | Designation | Source or reference | Identifiers | Additional information |
|---|---|---|---|---|
| strain, strain background (B. *thuringiensis*) | *Btk* SA-11 | Isolated from commercial product | Delfin | |
| strain, strain background (B. *thuringiensis*) | *Btk*$^{\Delta Cry}$ | BGSC | #4D22 | |
| strain, strain background (B. *thuringiensis*) | *Btk* producing Cry1Ac | BGSC | #4D4 | |
| strain, strain background (B. *thuringiensis*) | *Btk* producing Cry1Ab | This study | | Materials and Methods: Generation of *Btk*$^{Cry1Ab}$ |
| strain, strain background (*Escherichia coli*) | producing Cry1Ab | BGSC | #ECE54 | |
| strain, strain background (*Escherichia coli*) | producing Cry1Ac | BGSC | #ECE53 | |
| strain, strain background (*Escherichia coli*) | producing Cry2Aa | BGSC | #ECE126 | |
| genetic reagent (*D. melanogaster*) | WT canton S | https://bdsc.indiana.edu/ | #64349 | |
| genetic reagent (*D. melanogaster*) | *w; Sco/CyO; tub-GAL80$^{ts}$/ TM6b* | https://bdsc.indiana.edu/ M. Vidal | #7018 | |
| genetic reagent (*D. melanogaster*) | *w; tub-GAL80$^{ts}$; TM2/TM6b* | https://bdsc.indiana.edu/ | #7019 | |
| genetic reagent (*D. melanogaster*) | *w;; Dl-GAL4/TM6b* | S. Hou and X. Zeng; **Zeng et al., 2010** | | |
| genetic reagent (*D. melanogaster*) | *w; tub-GAL80$^{ts}$; Dl-GAL4 UAS-GFP/TM6b* | This study | | Can be obtained from Gallet's lab |
| genetic reagent (*D. melanogaster*) | *w; esg-GAL4$^{NP5130}$* | https://bdsc.indiana.edu/ N. Tapon | #67054 | |
| genetic reagent (*D. melanogaster*) | *w; esg-GAL4$^{NP5130}$ UAS-GFP* | N. Tapon; **Shaw et al., 2010** | | |
| genetic reagent (*D. melanogaster*) | *w; esg-GAL4$^{NP5130}$ UAS-GFP; tubGAL80$^{ts}$* | Y. Apidianakis; **Apidianakis et al., 2009** | | |
| genetic reagent (*D. melanogaster*) | *w; Su(H)GBE-GAL4, UAS-CD8::GFP* | M. Vidal; **Zeng et al., 2010** | | |
| genetic reagent (*D. melanogaster*) | *w; Su(H)GBE-GAL4/SM6; tub-GAL80$^{ts}$ UAS-GFP/TM6b* | This study | | Can be obtained from Gallet's lab |
| genetic reagent (*D. melanogaster*) | *w; myo1A-GAL4* | N. Tapon; **Shaw et al., 2010** | | |
| genetic reagent (*D. melanogaster*) | *w; myo1A-GAL4 UAS-GFP/ CyO* | Y. Apidianakis; **Apidianakis et al., 2009** | | |

*Appendix 1 Continued on next page*

*Appendix 1 Continued*

| Reagent type (species) or resource | Designation | Source or reference | Identifiers | Additional information |
|---|---|---|---|---|
| genetic reagent (*D. melanogaster*) | *w; UAS-GFP/TM3 Sb* | https://bdsc.indiana.edu/ | #5430 | |
| genetic reagent (*D. melanogaster*) | *w; UAS-GFP::CD8; UAS-H2B::RFP/TM2* | T. Reiff and M. Dominguez; *Antonello et al., 2015* | | |
| genetic reagent (*D. melanogaster*) | *w; UAS-CD8::GFP; UAS-H2B::RFP, tub-GAL80<sup>ts</sup>/TM2* | T. Reiff and M. Dominguez; *Antonello et al., 2015* | | |
| genetic reagent (*D. melanogaster*) | *w; esg-GAL4, UAS-CD8::GFP/ CyO; UAS-H2B::RFP, tub-GAL80ts/TM6b* | T. Reiff and M. Dominguez; *Antonello et al., 2015* | | |
| genetic reagent (*D. melanogaster*) | *w; UAS-CD8::GFP; Dl-GAL4, UAS-H2B::RFP/TM6b* | This study | | Can be obtained from Gallet's lab |
| genetic reagent (*D. melanogaster*) | *w;; UAS-GC3Ai<sup>G7S</sup> (UAS-Casp::GFP)* | M. Suzanne; *Schott et al., 2017* | | |
| genetic reagent (*D. melanogaster*) | *w; UAS-shg-R (DE-Cadherin)* | https://bdsc.indiana.edu/ | #58494 | |
| genetic reagent (*D. melanogaster*) | *w; UAS-connectin* | JP Boquete and B. Lemaitre; *Zhai et al., 2017* | | |
| genetic reagent (*D. melanogaster*) | *y w, shg::Tomato* | https://bdsc.indiana.edu/ | #58789. | |
| genetic reagent (*D. melanogaster*) | *w;UAS-p35* | Tony Ip; *Amcheslavsky et al., 2009* | | |
| genetic reagent (*D. melanogaster*) | *Dl-ReDDM (w/w; UAS-CD8::GFP/UAS-CD8::GFP; Dl-GAL4, UAS-H2B::RFP/UAS-H2B::RFP, tub-GAL80<sup>ts</sup>)* | This study | | Can be obtained from Gallet's lab |
| genetic reagent (*D. melanogaster*) | *esg-ReDDM (w/w<sup>+</sup>; esg-GAL4, UAS-CD8::GFP/+; UAS-H2B::RFP, tub-GAL80<sup>ts</sup>/+)* | This study | | Can be obtained from Gallet's lab |
| genetic reagent (*D. melanogaster*) | *Su(H)-ReDDM (w/w; Su(H)-GAL4/UAS-GFP::CD8; tub-GAL80<sup>ts</sup> UAS-GFP/UAS-H2B:RFP)* | This study | | Can be obtained from Gallet's lab |
| cell line (*D. melanogaster*) | *Drosophila melanogaster* Schneider 2 (S2) cells | S2-DGRC Stock 181 | RRID:CVCL_Z992 | |
| antibody | Mouse monoclonal anti-Armadillo (ß-catenin) antibody | DSHB | Cat# N27A1 RRID:AB_528089 | 1/50 |
| antibody | Mouse monoclonal anti-Connectin antibody | DSHB | Cat# Connectin C1.427, RRID:AB_1066083 | 1/200 |
| antibody | Mouse monoclonal anti-Prospero antibody | DSHB | Cat# MR1A RRID:AB_528440 | 1/200 |
| antibody | Rabbit polyclonal anti-Cleaved Caspase-3 (Asp175) antibody | Cell Signalling | Cat# 9661 RRID:AB_2341188 | 1/600 |

*Appendix 1 Continued on next page*

*Appendix 1 Continued*

| Reagent type (species) or resource | Designation | Source or reference | Identifiers | Additional information |
|---|---|---|---|---|
| antibody | Rabbit polyclonal anti-phospho-Histone H3 (Ser10) antibody | Millipore | Cat# 06–570 RRID:AB_31017 | 1/1000 |
| antibody | Rabbit polyclonal anti-Cry1A antibody | *Babin et al., 2020* | | WB: 1/7500 i |
| antibody | Mouse monoclonal anti-actin antibody (ACTN05, C4) antibody | Invitrogen | Thermo Fisher Scientific Cat# MA5-11866, RRID:AB_10985365 | WB: 1/2000 |
| antibody | Goat anti mouse IgG (H+L) secondary antibody, AlexaFluor-647 | Invitrogen | Molecular Probes Cat# A-21235, RRID:AB_2535804 | 1/500 |
| antibody | Goat polyclonal anti mouse IgG (H+L) secondary antibody, AlexaFluor-546 | Invitrogen | Molecular Probes Cat# A-11003, RRID:AB_141370 | 1/500 |
| antibody | Goat polyclonal anti-rabbit IgG (H+L) secondary antibody, AlexaFluor-647 | Invitrogen | Thermo Fisher Scientific Cat# A32733, RRID:AB_2633282 | 1/500 |
| antibody | Goat polyclonal anti-rabbit IgG (H+L) secondary antibody, AlexaFluor-546 | Invitrogen | Thermo Fisher Scientific Cat# A-11010, RRID:AB_253407 | 1/500 |
| recombinant DNA reagent | pUAST-DECadherintagged with GFP (DEFL) | *Oda and Tsukita, 1999* | | Materials and Methods: Cell aggregation assay |
| recombinant DNA reagent | pWA-Gal4 | Gift from L. Ruel | | |
| Software, algorithm | Image J | http://imagej.nih.gov | RRID:SCR_003070 | |
| Software, algorithm | Fiji | http://fiji.sc | RRID:SCR_002285 | |
| Software, algorithm | ZEN 2 (blue edition) | Zeiss | | |
| Software, algorithm | Photoshop CS2 | Adobe | | |
| Software, algorithm | GraphPad Software | GraphPad Prism | RRID:SCR_002798 | GraphPad Prism 7.0 |
| Chemical compound, drug | PBS 10 x | Euromedex | ET330 | |
| Chemical compound, drug | Formaldehyde 16% | Thermo Fisher Scientific | Cat# 28908 | |
| Chemical compound, drug | Fluoroshield-DAPI | Sigma | Cat# F6057 | |
| Chemical compound, drug | Tween 20 | VWR | Cat# 28829.296 | |
| Chemical compound, drug | Acrylamide/Bis-acrylamide | Sigma | Cat# A3699 | |
| Commercial assay, kit | Invitrogen MycoFluor Mycoplasma Detection Kit | Thermo Fisher Scientific | Cat# 10063202 | |

*Appendix 1 Continued on next page*

*Appendix 1 Continued*

| Reagent type (species) or resource | Designation | Source or reference | Identifiers | Additional information |
|---|---|---|---|---|
| Other | Amersham Hyperfilm | GE Healthcare | Cat# 28906837 | Commercial product |
| Other | Bovin Serum Albumin | Sigma | Cat# A9647 | Commercial product |
| Other | Schneider's insect medium | Sigma-Aldrich | Cat# S0146 | Commercial product |
| Other | TransIT–2020 | Mirus Bio | Cat# MIR5400 | Commercial product |
| other | Zeiss Axioplan Z1 with Apotome 2 microscope | Zeiss | | Microscope |
| other | Zeiss Confocal LSM 810 | Zeiss | | Microscope |

