## [Editor Report]

The microbial pathogen Bacillus thuringiensis subsp. kurstaki (Btk) and its Cry toxins are used extensively to kill lepidopteran crop pests. Although Btk is not lethal to *Drosophila*, Jneid et al. present convincing evidence that Btk's Cry1A toxins disrupt *Drosophila* intestinal homeostasis by inducing enterocyte death, activating stem cell divisions, and promoting excess enteroendocrine differentiation via weakening of adherens junctions between stem cells and terminal daughter cells. These important findings raise the possibility that Btk and Cry-based insecticides may alter the intestinal lining of non-targeted animal species.

---

## [Decision Letter]

**Decision letter after peer review:**

Thank you for submitting your article "Bacillus thuringiensis toxins divert progenitor cells toward enteroendocrine fate by decreasing cell adhesion with intestinal stem cells" for consideration by *eLife*. Your article has been reviewed by 3 peer reviewers, and the evaluation has been overseen by a Reviewing Editor and Arturo Casadevall as the Senior Editor. The following individuals involved in review of your submission have agreed to reveal their identity: Karen Bellec (Reviewer #1); Young Kwon (Reviewer #2); Yiorgos Apidianakis (Reviewer #3).

Essential revisions:

1. In several instances (e.g. Figure 1D, Figure 2 C and D), conclusions about changes in numbers of cells are based upon plots that count number of cells per unit area. However, cell number per unit area is a measure of cell density, not overall cell number. Please either provide total cell counts for a defined midgut region (e.g. R4), or else modify these conclusions. Alternatively, the ratio of particular cell types to total cells could be helpful as a measure of the relative abundance of different cell populations. In addition, could images that illustrate overall organ morphology be included? Such information would provide a sense of the relative sizes of the organs.

2. Please provide evidence to support the use of the Δ-ReDDM system as a readout of symmetric versus asymmetric fate outcomes. (a) Does degradation of the ReDDM GFP signal occur in the same time frame as activation of the enteroblast-specific SuH marker? (For instance, is ReDDM GFP signal absent from SuH-lacZ cells in both control and experimental conditions?) (b) Cell contact was used to classify two progenitor cells as siblings from the same mother stem cell division. Please either justify this claim or else revise the interpretation of these data. (c) Can you respond to Reviewer 3's question about co-expression of two GFP transgenes in the Δ-ReDDM genotypes?

3. Please address the Reviewers' requests for improved data presentation. (individual values in graphs, scale bars in images, loading controls for Westerns, etc). In addition, please quantify the correlation between Tomato::DE-Cad and EEP markers (described in lines 283-284).

4. The Reviewers raised questions about visibility and interpretation of the Arm, E-cad, and caspase signals in multiple figure panels. In cases where improved visibility would be helpful, please present split channel images. In cases where alternative interpretations are raised (e.g., could Arm levels be affected by proliferation rate?), please evaluate these alternatives either through new data or textual discussion.

5. Please clarify the methodological details requested by the Reviewers. (a) Better descriptions are needed of how quantitative analyses were performed. (b) In some cases, the criteria used to identify particular cell types or populations were ambiguous (e.g. new cells versus old cells, sibling cells versus neighbor cells, diploid enteroblasts identified via nuclear size), (c) To avert reader confusion, might you consider referring to enteroendocrine cells by the standard abbreviation EE, rather than EEC?

*Reviewer #1 (Recommendations for the authors):*

1) Data representation

Different figures should be modified to improve the representation of the data all along the manuscript:

– Authors should plot individual values in the graphs shown all along the manuscript instead of using histograms.

– Scale bars should be included both in the pictures and in the legends.

– Figure 5G-G': errors bars should put in another color. The scale of the graph could be changes to have a better overview of the distribution of the different points.

For each figure, the signification of the error bars (sd or sem, etc..) should be included in the legends and/or in the Material and Methods.

2) Experiments

– Loading controls of the Western blots are missing: authors should provide a control to indicate that the same amount of sample has been loaded in each condition by staining another protein. This is particularly true for the Western blots shown in Figure 6D and Figure 6E

– Figure S1A-S1B: How were the authors able to confirm that two cells came from the same mother cell? This question is particularly true for BtkSA11 where there is a lot of neighboring labelled cells. Details about the way to quantify should be added in the Material and Methods (quantification in a small area only? n=number of divisions analyzed?)

Same figure: The picture of Btk∆Cry seems to show lot of symmetric cell divisions giving rise to cells that are GFP+RFP+Pros- (=ISCs) but surprisingly, the Figure 1C does not show this increase.

– L181: Authors conclude: "Therefore, consistent with previous studies, the mode of ISC division appeared to reflect the degree of intestinal damage, the symmetric mode being preferentiality adopted when the damage was greater (i.e., when cell death occurred)." This mode of division seems to be also preferentially adopted when there is intestinal damage but no cell death, as in the case of Btk∆Cry.

*Reviewer #2 (Recommendations for the authors):*

Overall, the manuscript is in a good shape.

*Reviewer #3 (Recommendations for the authors):*

The described work is about assessing *Drosophila* midgut histopathology upon consumption of an entomopathogenic strain of B. thuringiensis and its Cry1A toxins, which are lethal to lepidoptera, but non-lethal to *Drosophila*. Thus, *Drosophila* is characterized a non-susceptible organism. The authors tested if this "non-susceptible host" is nevertheless histopathologically susceptible. They convincingly show that it is, because the mechanism of action of the Cry1A toxins on progenitor cell E-Cadherin is functionally (but not biochemically) revealed in flies and in *Drosophila* S2 cells.

To improve the manuscript, I suggest the following:

1. Add "*Drosophila* midgut" in the title, because generalizations need to be omitted.

2. Lines 89-92: reference to B. cereus should not add to the proof about Btk. So need to phrase this accordingly. Different species and different strains have different effects.

3. Figure 1A: how do Caspase 3 sensor and Caspase 3 antibody compare? While both are of a value, the authors measure the first, but show the latter. Why?

4. Dl-ReDDM experiments: The M and Ms read "w; tub-GAL80ts; Dl-GAL4 UAS-GFP/TM6b females were crossed to w; UAS-GFP::CD8; UAS-H2B::RFP/TM6 males". This means that the progeny express two different UAS-GFP transgenes, right? This issue applies also to ReDDM experiments with the other GAL4 lines. What is the perdurance of UAS-GFP in each case and in combination with UAS-GFP::CD8?

5. Line 224: Why the authors chose esg-ReDDM flies instead of Dl-ReDDM? The latter makes clear what ISC is, while esg based ReDDM confuses the distinction between ISC and EB.

6. Line 229: "EBs were GFP+ RFP+ DAPI+ with bigger nuclei". Why bigger? EBs are presumably diploid.

7. Figure S3: The terms "old cells" and "new cells" seems misleading, because by "old cells" the authors mean EEs of Su(H)-independent origin and by "new cells" EEs of Su(H)-dependent origin.

8. Lines 293-4: "the decrease in Tomato::DE-Cad 283 labelling between an ISC and its progenitor was correlated with the expression of EEP markers". The authors should measure the extend of this correlation. Also, neighboring esg+ cells do not equal "ISC and its progenitor".

9. The following statement is imprecise "we no longer observed an increase in the number of EECs (blue arrows in Figure 5A-D, and Figure 5E)". Comparing Figure 3F and Figure 5E, there is an increase in EEs in both cases, but the increase is more prominent in 3F.

10. I suggest replacing the term EEC(s) with EE(s) to abide to previous nomenclature and make it easier to distinguish from the term EEP(s).

11. Figure panels 5G and S5I are missing from the PDF available for this review.

12. Figure S2I ("old" and "new" pros+ cells are plotted together) refers to the anterior midgut only and to see what is going on in the posterior is unclear (information is not plotted). One needs to extrapolate maybe by comparing with Figure 3F ("old" and "new" pros+ cells are plotted separately).

13. Lines 363-4. To explain the discrepancy between the anterior and the posterior midgut in terms of EEs and their progenitors, the authors should resort to Tamamouna et al. 2020 (PMID: 32513656) and Marianes and Spradling 2013 (PMID: 23991285), where such gut compartmentalization and anterior vs. posterior mitosis and propensity for EE and ISC accumulation has been thoroughly discussed.

14. Line 492. What does "fortuitously mean? How was the selection process arranged? Was there a random mutagenesis involved?

[Editors’ note: further revisions were suggested prior to acceptance, as described below.]

Thank you for resubmitting your work entitled "*Bacillus thuringiensis* toxins divert progenitor cells toward enteroendocrine fate by decreasing cell adhesion with intestinal stem cells in *Drosophila* midgut" for further consideration by *eLife*. Your revised article has been evaluated by Arturo Casadevall (Senior Editor) and by Lucy O'Brien (Reviewing Editor), in consultation with the three original reviewers.

We all agreed that the revised manuscript addresses many, but not all, of the Essential Revision Items raised in the first round of review. In particular, concerns about cell density (Essential Revision Item #1) have been addressed, and the manuscript is much improved in terms of the data presentation and methodological detail (Essential Revision Items #3-5).

However, the overall consensus was that fundamental concerns still remain about the classification of division fate outcomes in the manuscript (Essential Revision Item #2). The authors were requested to perform two important controls to justify their classification: (1) examine whether degradation of the Δ-driven ReDDM GFP signal occurs in the same time frame as activation of an enteroblast-specific SuH marker, such as Su(H)-lacZ, and (2) justify their claim that progenitor cells that are in contact are sibling cells. As we explain in detail below ("Essential Revision Item #2 – Rationale"), the Response does not adequately address either of these points. Hence, we are not able to recommend the publication of the manuscript in its current form.

In light of the fact that fate outcomes are only one aspect of the study's overall conclusions, and that experiments to demonstrate division fate outcomes would represent substantial additional effort, we may consider publication if the authors were willing to revise their manuscript so as to refrain from drawing conclusions regarding symmetric and asymmetric fates. (It would be acceptable to include a speculative statement about fate outcomes, clearly indicated as such.) Without textual revisions that address these issues, or without new, direct evidence as originally requested, we are sorry that we are unable to move forward with this manuscript.

In addition, a few other items arose during the consultation; these can all be addressed by textual revision:

– Findings from prior studies described in lines 167-171 are described inaccurately (see explanation below), and two papers that contradict the authors' assertion were not cited.

– Other portions of the manuscript, such as the Figure 7 results, draw conclusions that are not supported by data. Careful revision of the study's results to avoid over-interpretation and to make clear distinctions between data-based conclusions and interesting, but speculative, remarks will also be necessary for a revised manuscript to be considered further.

– The title appears to be missing a "the". Two grammatically correct alternatives would be:

"Bacillus thuringiensis toxins divert progenitor cells toward enteroendocrine fate by decreasing cell adhesion with intestinal stem cells in the *Drosophila* midgut" or "Bacillus thuringiensis toxins divert progenitor cells toward enteroendocrine fate by decreasing cell adhesion with intestinal stem cells in *Drosophila*"

Essential Revision Item #2 – Rationale

(1) In terms of Point (1) above, the authors did not perform the requested experiment to examine whether degradation of the Δ-driven ReDDM GFP signal occurs in the same timeframe as activation of an enteroblast-specific SuH marker. Instead, they offer two alternative justifications for the validity of their current approach. However, as detailed below, neither of these justifications is valid.

The first justification the authors give for drawing conclusions about cell identity from ReDDM hinges on the half-life of GFP in the ReDDM system being less than 24 hours. The authors cite Antonello 2015 as evidence for the < 24-hour half-life, but a quantitative measurement of GFP half-life in the Antonello study could not be identified in either the main figures or the supplemental figures. (If this is mistaken, please point out where these data are found in the paper.) In addition, the authors also claim that since they examined tissues two days after ReDDM induction, the GFP transmitted to most daughter cells has begun to be degraded. This statement appears to assume that all cell divisions happened at the time that ReDDM was induced. However, this assumption is erroneous. Stem cells continue to cycle throughout the two-day post-induction period, and with fixed tissues, it cannot be ascertained when during the two-day period a division actually occurred. It is likely that some stem cell divisions occurred a short time prior to fixation.

The second justification that the authors provide is two images of DeltaGAL4-driven ReDDM showing that, following feeding with the Btk mutant, there is an increased number of GFP-/RFP+ cells compared to control guts. In both images, the authors designate the GFP-/RFP+ cells as enteroblasts and the GFP+/RFP+ cells as stem cells, and they conclude there are more enteroblasts following Btk treatment because there are more red cells. Without a positive marker of enteroblast fate, however, the authors' reasoning is neither scientifically nor logically valid. Crucially, enteroblasts are not defined by the absence of Δ (for instance, loss of Δ does not result in loss of stem cells, but rather results in excess stem cells and enteroendocrine cells -- Ohlstein 2007 Science). Instead, enteroblasts are defined by the activation of Notch targets such as the Su(H)GBE Notch reporter. Hence Notch activation and Δ downregulation are not necessarily synchronous. Rather, Δ downregulation may lag behind Notch activation, such that new enteroblasts transiently exist in a Δ-positive, Notch-activated state. Indeed, Zhai 2017 PLoS Geneticsand Tian 2022 EMBO J have found that under injury conditions there is a marked increase in Su(H)+ enteroblasts that are also Δ+.

(2) In terms of Point (2) above, the authors do not present data to support the claim that a GFP-/RFP+ cell adjacent to a GFP+/RFP+ cell is the division progeny of the GFP+/RFP+ cell, and they state that two days after induction is too short a time frame for enteroblasts to come into contact with a stem cell that is not its mother stem cell. By contrast, Martin 2018 (*eLife*), which analyzed stem cell divisions in live movies, found that approximately one-third of sibling cells separate from each other after birth. The duration of these movies was ~16 hours, which is much shorter than the 2-day period the authors used. In addition, Martin et al. observed that some of these sibling cells subsequently associated with other progenitor cells. These findings undermine the authors' assertion and underscore the importance of performing lineage tracing to deduce lineage relationships. While we appreciate that some other journals have previously published midgut studies that assume progenitor cell pairs are siblings, our view is that publication in *eLife* requires more rigorous evidence such as lineage tracing or direct monitoring of live divisions.

Finally, the statement that "upon environmental challenges, ISCs divide symmetrically" (Author's response and manuscript lines 167-171) is not supported by the literature. The deNavascues 2012, Goulas 2012, Perdigoto 2011, and Tian 2014 studies examine homeostasis only; hence, these results do not speak to what happens when there are environmental challenges. In addition, Goulas and Perdigoto do not explicitly examine asymmetric versus symmetric division fate outcomes. O'Brien 2011 examined feeding-induced growth, which is not an environmental challenge but rather a normal physiological mechanism; hence, these results also do not speak to what happens upon environmental challenges. Although Zhai 2017 speculates on symmetric fate outcomes in the discussion, no conclusions are drawn about the fate outcomes of cell pairs in the data presented in the Results section.

By contrast, there are two studies that did examine division fate outcomes in injured guts, and that were not cited in either the Response or the manuscript: Jin 2017 (Stem Cell Reports) and Tian 2017 (PNAS). Both studies used Twin-spot MARCM – the gold standard for symmetric/asymmetric division fate outcomes – and found that the majority of divisions still exhibit asymmetric fate outcomes following injury. Their injury conditions of *Pseudomonas* infection and bleomycin ingestion are arguably in a similar realm to the current study's Btk treatment. Their results contradict the authors' assertion that stem cells divide symmetrically following environmental challenges and raise the possibility the authors' method is inaccurate.

Additional remarks from individual reviewers follow below:

*Reviewer #1 (Recommendations for the authors):*

Thanks to the authors for addressing the questions raised in the previous reviews and for their detailed answers.

In the previous reviews, we highlighted the lack of some quantifications; controls, and information about the methods of analysis for some key experiments (Arm staining; rescue experiments..). The authors addressed all the questions when experimentally possible and modified the text accordingly. They also modified the graphs as suggested. Altogether, these modifications greatly improved the manuscript and reinforced the conclusions proposed by the authors.

*Reviewer #2 (Recommendations for the authors):*

The authors have appropriately addressed most of the comments. However, I have a concern regarding the results presented in Figure 7. The results raise interesting ideas about how toxins might reach ISCs. The presented experiments do not directly test the ideas. Therefore, the conclusions are not appropriately supported. I suggest the authors revise the text to clearly separate the conclusions drawn from the experimental data and their speculations. Alternatively, the whole data can be removed as those are not essential for the story.

*Reviewer #3 (Recommendations for the authors):*

Taking the authors' words (in their response to reviewers): "ISCs mostly divide asymmetrically while upon environmental challenges ISCs divide symmetrically (de Navascues et al., 2012; Goulas et al., 2012; O'Brien et al., 2011; Perdigoto et al., 2011; Tian and Jiang, 2014; Zhai et al., 2017)". Thus, being responsive to Bacillus thuringiensis infection and its Cry1A toxins does not equal B. thuringiensis being a hazard to *Drosophila* or other flies. Such treatments merely elicit a defense response against damaging agents, which are mild enough for flies to cope with (survive). I suggest the authors refrain from conclusions (should there be any) about B. thuringiensis being very hazardous to flies.

---

## [Author Response]

Essential revisions:1. In several instances (e.g. Figure 1D, Figure 2 C and D), conclusions about changes in numbers of cells are based upon plots that count number of cells per unit area. However, cell number per unit area is a measure of cell density, not overall cell number. Please either provide total cell counts for a defined midgut region (e.g. R4), or else modify these conclusions.

In our analysis, we generally took images in the middle of the R4 region (in general 2 pictures/gut) to avoid variations due to differences in R4 length or to mis determination of the R4 region boundaries. Then we performed our cell counting in a region of interest (ROI) of about 20.000µm² because this corresponds to the mean analyzable surface on a picture taken at the 40X magnification.

We provided more explanation in Material and Methods – section measurement, counting and statistical analysis lines 729-748. For more clarity, we also added images in Figure 1—figure supplement 1A delimiting the R4 region in the different feeding conditions.

Alternatively, the ratio of particular cell types to total cells could be helpful as a measure of the relative abundance of different cell populations.

We thank the reviewer for this measurement suggestion. We have now added all ratio in Figure 1—figure supplement 2.

We have also modified the text all along the manuscript to integrate this method of analysis. Noteworthy when both cell density and cell ratio changes (increased or decreased) for a given cell type, we assumed that this corresponded to a global changes in the so-called cell type number.

In the figure legends, we also replaced "cell number" by "cell density".

In addition, could images that illustrate overall organ morphology be included? Such information would provide a sense of the relative sizes of the organs.

We have added whole posterior midgut views at 5X magnification in Figure 1—figure supplement 1A, and we have modified the text accordingly (lines 144-145).

2. Please provide evidence to support the use of the Δ-ReDDM system as a readout of symmetric versus asymmetric fate outcomes. (a) Does degradation of the ReDDM GFP signal occur in the same time frame as activation of the enteroblast-specific SuH marker? (For instance, is ReDDM GFP signal absent from SuH-lacZ cells in both control and experimental conditions?)

It is widely admitted that Dl-Gal4 driver is only expressed in ISCs and enteroendocrine precursors (EEPs), and never reported to be expressed in Enteroblasts (Su(H)+ cells) (Biteau and Jasper, 2014; Guo and Ohlstein, 2015; Joly and Rousset, 2020; Zeng and Hou, 2015).

In the ReDDM system, the half-life of the GFP is less than 24h while the H2B::RFP we used is at least 2 weeks (Antonello et al., 2015). So even if GFP expressed in ISCs might be transmitted to EB daughter cells, the GFP intensity in EBs would be very week due to the short half-life of GFP. Since we took pictures 2 days after ReDDM induction (see Figure 1 —figure supplement 1D), we assume that the GFP transmitted to most of EB daughter cells has begun to fade. Indeed, increasing the intensity using Zeiss or Photoshop software allows to distinguish a weak GFP staining in the putative EBs. However, we consider this faint staining as background.

Author response image 1 shows a Dl-ReDMM midgut fed with water (Control), 2 days after temperature shift at 29°C (i.e. the induction of GFP and RFP). We have intentionally saturated the fluorescence intensity.

1st: GFP+/RFP+ cells (corresponding to ISCs) are mainly isolated cells.

2nd: RFP+/GFP- small nucleated cells (that are likely EBs) are only rarely labeled by a weak green florescence even when GFP intensity is intentionally increased.

**Author response image 1. sa2fig1:** 

Author response image 2 is an example of a Dl-ReDMM experiment using flies fed with Btk^ΔCry^ that stimulate the production of EBs. Many RFP+/GFP- are produced and the GFP is barely detectable in a few EBs even when we forced the intensity of the fluorescence.Therefore, we believe in our conclusions based on the use of the Dl-ReDDM tool.

(b) Cell contact was used to classify two progenitor cells as siblings from the same mother stem cell division. Please either justify this claim or else revise the interpretation of these data.

In control conditions, EEs are rarely found in close contact with an ISC (see all our control condition images in our figures and figure supplements). Therefore, an EEP GFP+/RFP+/Pros+ in direct contact with an ISC GFP+/RFP+ has most likely inherited the GFP from this neighboring ISC. A GFP-/RFP+ cell with a small nucleus (an EB) contacting a GFP+/RFP+ ISC is likely to be its daughter cell. Although we can imagine that a GFP-/RFP+ EB contacting an GFP+/RFP+ ISC could arise from a distant GFP+/RFP+ ISC, we took images 2 days after induction of the ReDDM system limiting the time frame allowing a distant EB to come in close contact to an ISC which is not its mother cells. This is explained in the text lines 172-187.

Moreover using these criteria, we found ratio of symmetric/asymmetric divisions close to what it has been previously described: ISCs mostly divide asymmetrically while upon environmental challenges ISC divide symmetrically (de Navascues et al., 2012; Goulas et al., 2012; O'Brien et al., 2011; Perdigoto et al., 2011) (Tian and Jiang, 2014; Zhai et al., 2017). These references have been added line 184-186.

We have also added a sentence in the Material and Methods section lines 627-630.

(c) Can you respond to Reviewer 3's question about co-expression of two GFP transgenes in the Δ-ReDDM genotypes?

Indeed, we used two copies of GFP in F1 flies because Dl-Gal4 driver was not strongly expressed. When Dl-Gal4 drives the expression of one GFP allele, the GFP is weakly expressed (see the expression of the CD8::GFP in Author response image 3, red labelling is Prospero). Therefore, this led us to use two GFP copies in order to label most mother cells in the Dl-ReDDM experiments. In addition, if the GFP was weakly detectable while the RFP was strongly expressed, these cells would have been considered as progenies and not as mother cells. Of note, we also combined the membrane tethered (GFP::CD8) and a nuclear GFP to increase the probability to detect the cells.

**Author response image 3. sa2fig3:** 

Similarly, we expressed two copies of the GFP in the Su(H)-ReDDM experiments to track most EBs (mother cells). We also confirmed that Su(H)-Gal4 expression was restricted to EBs with no leak in other cell type since the progenies in the Su(H)-ReDDM experiments were only ECs (RFP+/GFP).

3. Please address the Reviewers' requests for improved data presentation. (individual values in graphs, scale bars in images, loading controls for Westerns, etc). In addition, please quantify the correlation between Tomato::DE-Cad and EEP markers (described in lines 283-284).

i) We have now added the individual values in the graphs of figure 1 (and supplement), figure 2 (and supplement), figure 3—figure supplement 1, figure 5 (and supplement) and figure 6.

ii) We have added the scale bars on figures and the magnification in the legends.

iii) We have added the loading control for Western Blots.

For figure 6D, we cannot use classical loading controls such as actin since Bt does not have actins. So we used Coomassie blue staining of the acrylamide gel. The gel is presented in Figure 6 —figure supplement 1J.

For figure 6E: Loading controls have been realized after blotting membranes with the monoclonal Actin antibody from Invitrogen (ACTN05, C4). As we used detergent-free cell lysis buffer, actin was mainly found in the insoluble fraction (Orlando et al., 2001). This is now explained in the text lines 389-392, in the Material and Methods lines 717-718 and in Figure 6E legend.

Moreover, we always standardized the amount of proteins we have deposited in the each well of the gels: 20µg of midgut proteins and 2.10^4^ CFU of spores (right part of the gel in figure 6D) were deposited. This is explained lines 710-7119.

iv) To include this quantification, we have rebuilt the related figures. Tomato::DE-Cad images and quantification are now presented in figure 4. Arm staining and quantification have been moved in figure 4—figure supplement 1A-E. We have also illustrated our method of quantification in figure 4—figure supplement 1F-G. Finally, we have included the measurement value in Table 1. We have edited the text and accordingly lines 285-297. In the Material and Methods, we added a section to explain better the way we proceeded to perform this analysis (lines 736-742).

4. The Reviewers raised questions about visibility and interpretation of the Arm, E-cad, and caspase signals in multiple figure panels. In cases where improved visibility would be helpful, please present split channel images. In cases where alternative interpretations are raised (e.g., could Arm levels be affected by proliferation rate?), please evaluate these alternatives either through new data or textual discussion.

i) We have now added a new figure 4 (the old figure 4 has been combined with the Figure 4—figure supplement 1), and we drew table 1 to facilitate the interpretation of our quantification and improve the visibility. Details of the quantification methods have been added lines 736-742. See also our response to reviewer 1 point 2.

ii) We have now put the *myo1A>Casp::GFP* labelling in figure 1A. The anti-cleaved Caspase staining has been moved in Figure 1—figure supplement 1B with split channels. We have also adapted the text to these modifications.

iii) See our response to Reviewer 1 point 2. We have also improved the Arm visibility in Figure 5—figure supplement 1 A-D' (previously Figure S4A-D).

5. Please clarify the methodological details requested by the Reviewers. (a) Better descriptions are needed of how quantitative analyses were performed.

We provided the reviewers a detailed explanation of our quantitative analyses and added new texts and sections in the Material and Methods (lines 627-630 and 727-742).

(b) In some cases, the criteria used to identify particular cell types or populations were ambiguous e.g. new cells versus old cells, sibling cells versus neighbor cells, diploid enteroblasts identified via nuclear size,

We addressed these issues accordingly: see our responses to the Editor point 2(b), to the Reviewer 1 point 2 and to the Reviewer 3 points 6 and 7.

(c) To avert reader confusion, might you consider referring to enteroendocrine cells by the standard abbreviation EE, rather than EEC?

This have been corrected.

Reviewer #1 (Recommendations for the authors):1) Data representationDifferent figures should be modified to improve the representation of the data all along the manuscript:– Authors should plot individual values in the graphs shown all along the manuscript instead of using histograms.

Done for graphs excepted for those that analyzed the old versus new cells. Putting all individuals value on the graph makes the graph harder to read. So we kept the initial presentation.

– Scale bars should be included both in the pictures and in the legends.

Scale bar added.

– Figure 5G-G': errors bars should put in another color. The scale of the graph could be changes to have a better overview of the distribution of the different points.

These were not the error bars but the averages. We have put the bars in black. We tried to change the scale but we were losing the highest points. We also tried different kinds of graphs and we found that the ones presented in the figures 5 were the most representative and easy to read.

For each figure, the signification of the error bars (sd or sem, etc..) should be included in the legends and/or in the Material and Methods.

This has been added to Methods lines 747-748.

2) Experiments– Loading controls of the Western blots are missing: authors should provide a control to indicate that the same amount of sample has been loaded in each condition by staining another protein. This is particularly true for the Western blots shown in Figure 6D and Figure 6E

See our response to the "Essential Revisions (for the authors) point 3. To summarize, we have added the loading control. Moreover, we always standardized the amount of proteins we deposited in each well of the gels: 20µg for midgut proteins and 2.10^4^ CFU of spores (right part of the gel) were deposited. This is explained lines 710-712. Noteworthy, anti-Actin cannot be used for controlling the loading charge of Bt spores that do not have actins.

– Figure S1A-S1B: How were the authors able to confirm that two cells came from the same mother cell? This question is particularly true for BtkSA11 where there is a lot of neighboring labelled cells. Details about the way to quantify should be added in the Material and Methods (quantification in a small area only? n=number of divisions analyzed?)

We kindly refer the reviewer to our response to the "Essential Revisions, point 2b". We have added the explanations in the Material and Methods lines 627-630 and 725-748.

Indeed, n correspond to the number of samples analyzed. We have now specify to what n corresponds in the figure legends (intestines/40X images, cells or cell pairs).

Same figure: The picture of Btk∆Cry seems to show lot of symmetric cell divisions giving rise to cells that are GFP+RFP+Pros- (=ISCs) but surprisingly, the Figure 1C does not show this increase.

Both genotypes are different. In Figure 1C, Dl-Gal4 UAS-GFP/+ is used at 25°C. In Figure S1A-S1B (now Figure 1—figure supplement 1C-D), the genotype is Dl-ReDDM with the shift from 18°C to 29°C. Moreover, we noticed that the induction of the ReDDM system induces a stress (shifting flies from 18°C to 29°C) increasing the number of Esg+ cells and probably enhancing the symmetric division, giving rise to two ISCs.

Importantly, there are also EB-EB symmetric divisions (visible on the Figure 1—figure supplement 2D) that buffers the increase in ISCs in the Dl-Gal4 UAS/+ genotype (Figure 1C). Whether the divisions give rise to 2 EBs or 2 ISCs, this does not change the rate of symmetric vs asymmetric divisions presented in Figure 1 —figure supplement 1C.

– L181: Authors conclude: "Therefore, consistent with previous studies, the mode of ISC division appeared to reflect the degree of intestinal damage, the symmetric mode being preferentiality adopted when the damage was greater (i.e., when cell death occurred)." This mode of division seems to be also preferentially adopted when there is intestinal damage but no cell death, as in the case of Btk∆Cry.

We apologizes that our sentence was not well formulated. "Damage" does not means systematically "cell death". Cells can be damaged without resulting in cell death we already demonstrated this in (Loudhaief et al., 2017). So we have removed "(i.e., when cell death occurred)" line 187.

Reviewer #3 (Recommendations for the authors):The described work is about assessing *Drosophila* midgut histopathology upon consumption of an entomopathogenic strain of B. thuringiensis and its Cry1A toxins, which are lethal to lepidoptera, but non-lethal to *Drosophila*. Thus, *Drosophila* is characterized a non-susceptible organism. The authors tested if this "non-susceptible host" is nevertheless histopathologically susceptible. They convincingly show that it is, because the mechanism of action of the Cry1A toxins on progenitor cell E-Cadherin is functionally (but not biochemically) revealed in flies and in *Drosophila* S2 cells.

We thank the reviewer for appreciating our findings and for all his constructive comments.

To improve the manuscript, I suggest the following:1. Add "*Drosophila* midgut" in the title, because generalizations need to be omitted.

This has been fixed.

2. Lines 89-92: reference to B. cereus should not add to the proof about Btk. So need to phrase this accordingly. Different species and different strains have different effects.

We have rephrased the sentence as requested.

3. Figure 1A: how do Caspase 3 sensor and Caspase 3 antibody compare? While both are of a value, the authors measure the first, but show the latter. Why?

We have now modified and put in figure 1A the images corresponding to Caspase 3 sensor. The anti-cleaved-Casp3 staining has been moved in figure 1—figure supplement 1B.

4. Dl-ReDDM experiments: The M and Ms read "w; tub-GAL80ts; Dl-GAL4 UAS-GFP/TM6b females were crossed to w; UAS-GFP::CD8; UAS-H2B::RFP/TM6 males". This means that the progeny express two different UAS-GFP transgenes, right? This issue applies also to ReDDM experiments with the other GAL4 lines. What is the perdurance of UAS-GFP in each case and in combination with UAS-GFP::CD8?

We kindly refer the reviewer to our response to Editors' Essential Revisions point 2c.

5. Line 224: Why the authors chose esg-ReDDM flies instead of Dl-ReDDM? The latter makes clear what ISC is, while esg based ReDDM confuses the distinction between ISC and EB.

This choice was made based on technical reasons. Dl-ReDDM fly strain is very difficult to amplify and maintain. To perform several repetitions with high number of individuals, we decided to use esg-ReDDM to be sure to get enough F1 flies of the good genotypes. However, we carried out the Su(H)-ReDDM control to check that EBs did not transdifferentiate into EEs.

6. Line 229: "EBs were GFP+ RFP+ DAPI+ with bigger nuclei". Why bigger? EBs are presumably diploid.

We apologize for this mistake. Not "bigger nuclei" but "small". This has been corrected.

7. Figure S3: The terms "old cells" and "new cells" seems misleading, because by "old cells" the authors mean EEs of Su(H)-independent origin and by "new cells" EEs of Su(H)-dependent origin.

Now this is the Figure 3—figure supplement 1A-D. This is a great remark.

By new cells, we mean cells that have now appeared after the induction of the Su(H)-ReDDM system and that express the RFP. If cells express the RFP, this means that these RFP+/GFP- cells arise from Su(H) mother cells. This also mean that these RFP+/GFP- cells did not exist before the induction of the ReDDM system. Therefore, we consider these RFP+/GFP- cells as "newborn cells" compared to cells already present before the ReDDM induction. We have now added the counting of ECs in this *Su(H)-ReDDM* experiments (Figure 3—figure supplement 1D). In this latter experiment we do observe new enterocytes (RFP+/GFP-). Therefore, we clearly demonstrate that EBs can only give birth to "new" ECs and not to "new" EEs. We confirm the previous observation by Zeng and Hou (Zeng and Hou, 2015) showing that EEs never develop from Su(H)+ EBs. We modified the text lines 252-253.

8. Lines 293-4: "the decrease in Tomato::DE-Cad 283 labelling between an ISC and its progenitor was correlated with the expression of EEP markers". The authors should measure the extend of this correlation. Also, neighboring esg+ cells do not equal "ISC and its progenitor".

We have now added the quantification in figure 4E and Table 1 to highlight this correlation. We have also improved the whole Figure 4 and added details for the quantification in Figure 4—figure supplement 1F-G.

We also adapted the text (lines 285-299) to fine-tune our interpretation in light of this new analysis. Indeed, what we have now defined as "mild" adherens junction intensity is between the ratio 1.4 and 1.6 instead of the previous ratio (1.3 to 1.6), because we observed most of the EEP progenitors arising from cell displaying a junction intensity with their mother cells below the 1.4 ratio (see Table 1). The material and Methods have also been implemented (lines 736-742)*.*

9. The following statement is imprecise "we no longer observed an increase in the number of EECs (blue arrows in Figure 5A-D, and Figure 5E)". Comparing Figure 3F and Figure 5E, there is an increase in EEs in both cases, but the increase is more prominent in 3F.

The reviewer comment is right. There is a partial rescue in the increase of EEC when we overexpressed the DE-Cadherin. So, we have modified the text lines 308-310.

10. I suggest replacing the term EEC(s) with EE(s) to abide to previous nomenclature and make it easier to distinguish from the term EEP(s).

This has been fixed.

11. Figure panels 5G and S5I are missing from the PDF available for this review.

Sorry. This was a bug during the PDF building.

12. Figure S2I ("old" and "new" pros+ cells are plotted together) refers to the anterior midgut only and to see what is going on in the posterior is unclear (information is not plotted). One needs to extrapolate maybe by comparing with Figure 3F ("old" and "new" pros+ cells are plotted separately).

It is now added in the new Figure 2—figure supplement 2I to easily compare what is happening.

13. Lines 363-4. To explain the discrepancy between the anterior and the posterior midgut in terms of EEs and their progenitors, the authors should resort to Tamamouna et al. 2020 (PMID: 32513656) and Marianes and Spradling 2013 (PMID: 23991285), where such gut compartmentalization and anterior vs. posterior mitosis and propensity for EE and ISC accumulation has been thoroughly discussed.

We have now integrated these observations and these references lines 382-385.

14. Line 492. What does "fortuitously mean? How was the selection process arranged? Was there a random mutagenesis involved?

Fortuitously means by chance, since this experiment was originally carried out with the aim of generating a mutant depleted for the all cry genes, carried by distinct plasmids. After analysis of the sequencing data, we identified this partial mutant (Btk*^Cry1Ab^*) that we used for further analysis. The selection process was therefore arranged at 2 levels: (i) by selection of the crystal-negative phenotype, and (ii) after analysis of the genomic sequences.

The part (lines 540 to 550) were modified for more clarity.

[Editors’ note: further revisions were suggested prior to acceptance, as described below.]

In addition, a few other items arose during the consultation; these can all be addressed by textual revision:– Findings from prior studies described in lines 167-171 are described inaccurately (see explanation below), and two papers that contradict the authors' assertion were not cited.– Other portions of the manuscript, such as the Figure 7 results, draw conclusions that are not supported by data. Careful revision of the study's results to avoid over-interpretation and to make clear distinctions between data-based conclusions and interesting, but speculative, remarks will also be necessary for a revised manuscript to be considered further.– The title appears to be missing a "the". Two grammatically correct alternatives would be:"Bacillus thuringiensis toxins divert progenitor cells toward enteroendocrine fate by decreasing cell adhesion with intestinal stem cells in the *Drosophila* midgut" or "Bacillus thuringiensis toxins divert progenitor cells toward enteroendocrine fate by decreasing cell adhesion with intestinal stem cells in *Drosophila*"

This has been corrected.

Essential Revision Item #2 – Rationale(1) In terms of Point (1) above, the authors did not perform the requested experiment to examine whether degradation of the Δ-driven ReDDM GFP signal occurs in the same timeframe as activation of an enteroblast-specific SuH marker. Instead, they offer two alternative justifications for the validity of their current approach. However, as detailed below, neither of these justifications is valid.The first justification the authors give for drawing conclusions about cell identity from ReDDM hinges on the half-life of GFP in the ReDDM system being less than 24 hours. The authors cite Antonello 2015 as evidence for the < 24-hour half-life, but a quantitative measurement of GFP half-life in the Antonello study could not be identified in either the main figures or the supplemental figures. (If this is mistaken, please point out where these data are found in the paper.) In addition, the authors also claim that since they examined tissues two days after ReDDM induction, the GFP transmitted to most daughter cells has begun to be degraded. This statement appears to assume that all cell divisions happened at the time that ReDDM was induced. However, this assumption is erroneous. Stem cells continue to cycle throughout the two-day post-induction period, and with fixed tissues, it cannot be ascertained when during the two-day period a division actually occurred. It is likely that some stem cell divisions occurred a short time prior to fixation.The second justification that the authors provide is two images of DeltaGAL4-driven ReDDM showing that, following feeding with the Btk mutant, there is an increased number of GFP-/RFP+ cells compared to control guts. In both images, the authors designate the GFP-/RFP+ cells as enteroblasts and the GFP+/RFP+ cells as stem cells, and they conclude there are more enteroblasts following Btk treatment because there are more red cells. Without a positive marker of enteroblast fate, however, the authors' reasoning is neither scientifically nor logically valid. Crucially, enteroblasts are not defined by the absence of Δ (for instance, loss of Δ does not result in loss of stem cells, but rather results in excess stem cells and enteroendocrine cells -- Ohlstein 2007 Science). Instead, enteroblasts are defined by the activation of Notch targets such as the Su(H)GBE Notch reporter. Hence Notch activation and Δ downregulation are not necessarily synchronous. Rather, Δ downregulation may lag behind Notch activation, such that new enteroblasts transiently exist in a Δ-positive, Notch-activated state. Indeed, Zhai 2017 PLoS Geneticsand Tian 2022 EMBO J have found that under injury conditions there is a marked increase in Su(H)+ enteroblasts that are also Δ+.(2) In terms of Point (2) above, the authors do not present data to support the claim that a GFP-/RFP+ cell adjacent to a GFP+/RFP+ cell is the division progeny of the GFP+/RFP+ cell, and they state that two days after induction is too short a time frame for enteroblasts to come into contact with a stem cell that is not its mother stem cell. By contrast, Martin 2018 (eLife), which analyzed stem cell divisions in live movies, found that approximately one-third of sibling cells separate from each other after birth. The duration of these movies was ~16 hours, which is much shorter than the 2-day period the authors used. In addition, Martin et al. observed that some of these sibling cells subsequently associated with other progenitor cells. These findings undermine the authors' assertion and underscore the importance of performing lineage tracing to deduce lineage relationships. While we appreciate that some other journals have previously published midgut studies that assume progenitor cell pairs are siblings, our view is that publication in eLife requires more rigorous evidence such as lineage tracing or direct monitoring of live divisions.Finally, the statement that "upon environmental challenges, ISCs divide symmetrically" (Author's response and manuscript lines 167-171) is not supported by the literature. The deNavascues 2012, Goulas 2012, Perdigoto 2011, and Tian 2014 studies examine homeostasis only; hence, these results do not speak to what happens when there are environmental challenges. In addition, Goulas and Perdigoto do not explicitly examine asymmetric versus symmetric division fate outcomes. O'Brien 2011 examined feeding-induced growth, which is not an environmental challenge but rather a normal physiological mechanism; hence, these results also do not speak to what happens upon environmental challenges. Although Zhai 2017 speculates on symmetric fate outcomes in the discussion, no conclusions are drawn about the fate outcomes of cell pairs in the data presented in the Results section.By contrast, there are two studies that did examine division fate outcomes in injured guts, and that were not cited in either the Response or the manuscript: Jin 2017 (Stem Cell Reports) and Tian 2017 (PNAS). Both studies used Twin-spot MARCM – the gold standard for symmetric/asymmetric division fate outcomes – and found that the majority of divisions still exhibit asymmetric fate outcomes following injury. Their injury conditions of Pseudomonas infection and bleomycin ingestion are arguably in a similar realm to the current study's Btk treatment. Their results contradict the authors' assertion that stem cells divide symmetrically following environmental challenges and raise the possibility the authors' method is inaccurate.

We are grateful to the editors for the exhaustive bibliographic analysis they provided. We are agreeing that the methods we used cannot provide solid conclusions on the mode of ISC division upon Bt/toxin ingestion and more experiments are required. Since the mode of ISC division is not the focus of our work and is not linked to the main message of our publication, we followed the editor’ suggestion and therefore modified the text by removing all the sentences referring to the mode of ISC division. We hope the changes we have made are satisfactory now and have taken into consideration all the points raised by the reviewers and editors regarding the division fate outcomes upon Bt/toxin ingestion.

We have modified the text lines 48-49, 110, 161-174, 242 and 610. We removed the graph related to the mode of ISC division in the Figure 1 —figure supplement 1. The figure legend has been corrected accordingly lines 1201-1206.

Additional remarks from individual reviewers follow below:Reviewer #1 (Recommendations for the authors):Thanks to the authors for addressing the questions raised in the previous reviews and for their detailed answers.In the previous reviews, we highlighted the lack of some quantifications; controls, and information about the methods of analysis for some key experiments (Arm staining; rescue experiments..). The authors addressed all the questions when experimentally possible and modified the text accordingly. They also modified the graphs as suggested. Altogether, these modifications greatly improved the manuscript and reinforced the conclusions proposed by the authors.

Thank you very much for your suggestions that have increased the quality of our results.

Reviewer #2 (Recommendations for the authors):The authors have appropriately addressed most of the comments. However, I have a concern regarding the results presented in Figure 7. The results raise interesting ideas about how toxins might reach ISCs. The presented experiments do not directly test the ideas. Therefore, the conclusions are not appropriately supported. I suggest the authors revise the text to clearly separate the conclusions drawn from the experimental data and their speculations. Alternatively, the whole data can be removed as those are not essential for the story.

We have included these experiments and data in the first revised version to address the question raised by reviewer #2 during the first round of revision. According to the reviewer's new issue, we have rewritten this part (Lines 387-411) to better describe our results and moderate our conclusions.

Reviewer #3 (Recommendations for the authors):Taking the authors' words (in their response to reviewers): "ISCs mostly divide asymmetrically while upon environmental challenges ISCs divide symmetrically (de Navascues et al., 2012; Goulas et al., 2012; O'Brien et al., 2011; Perdigoto et al., 2011; Tian and Jiang, 2014; Zhai et al., 2017)". Thus, being responsive to Bacillus thuringiensis infection and its Cry1A toxins does not equal B. thuringiensis being a hazard to *Drosophila* or other flies. Such treatments merely elicit a defense response against damaging agents, which are mild enough for flies to cope with (survive). I suggest the authors refrain from conclusions (should there be any) about B. thuringiensis being very hazardous to flies.

We agree that Bt is not strongly hazardous to flies. Indeed, we have already demonstrated that Bt is a weak opportunistic bacteria causing moderate damage in the *Drosophila* intestine (Loudhaief et al., 2017). This is also what we mentioned clearly in this manuscript, lines 178-182. Of note, we have verified the text and we didn’t find the word hazardous in any section.

In our response to reviewers, we used the term "environmental challenge" (not used in the manuscript) to point to an environmental stimulus (not necessarily a strong aggression) that is perceived by the intestine as potentially harmful, requiring a local cellular response. But for sure, *Bt* is an opportunistic bacterium weakly virulent for *Drosophila* as well as for human since the main symptom is diarrhea.